# Rethinking the Spatiotemporal Distribution for High-Fidelity Parallel ANN-to-SNN Conversion

## Abstract

Spiking Neural Networks (SNNs) have attracted increasing attention for their low power consumption and their support for low-latency, potentially adaptive-timestep inference on neuromorphic hardware. Among existing approaches, ANN-to-SNN conversion is one of the most effective ways to obtain deep SNNs with accuracy comparable to traditional ANNs, and recent work has even extended it to *parallel* conversion, where the full spike train is emitted in a single pass. Despite this promise, we find that ANN-to-SNN parallel conversion suffers from severe performance degradation at ultra-low timesteps ($T \leq 4$), limiting its practical use. In this work, we analyze the source of this performance gap and demonstrate that it originates from assumptions in the standard quantization–clip–floor–shift (QCFS) formulation, which, under the one-shot firing rule, introduces a timestep-dependent bias. To overcome this, we propose a *Distribution-Aware (DA) Parallel Calibration* that corrects spatiotemporal mismatches while leaving the backbone and firing rule unchanged. Our method consists of two stages: (1) spatial recalibration, which brings the backbone into a DA-QCFS form and recalibrates normalization layers so that their statistics match the spike-domain distributions required by subsequent parallel conversion, and (2) temporal correction, which learns a per-channel bias over the time-collapsed membrane potential to offset timestep-dependent errors. On ImageNet-1k, using **off-the-shelf** ResNet backbones without any QCFS-specific retraining, our approach boosts top-1 accuracy from $\mathbf{25.20}\% \rightarrow \mathbf{62.28}\%$ for ResNet-18 at $T = 4$ and from $\mathbf{50.67}\% \rightarrow \mathbf{68.23}\%$ for ResNet-34 at $T = 8$, and it yields similarly large gains when starting from QCFS/DA-QCFS-based backbones (e.g., ResNet-34 improves from $\mathbf{42.45}\% \rightarrow \mathbf{72.35}\%$ at $T = 2$).

## 1 Introduction

Spiking Neural Networks (SNNs) are widely regarded as a promising paradigm for energy-efficient computation, thanks to their sparse, event-driven communication that mirrors biological neurons. Among deployment strategies, *ANN-to-SNN conversion* has emerged as one of the most effective techniques, as it leverages the maturity of deep learning frameworks to achieve strong performance without the cost of direct SNN training (Bu et al., 2022). Within this paradigm, *parallel conversion* (Hao et al., 2025) is particularly attractive: by generating the entire spike train in a single pass, it enables inference with constant time complexity, regardless of the number of timesteps ($T$). This property makes ultra-low-latency inference (e.g., $T \leq 4$) theoretically attainable.

In practice, however, *parallel conversion* strategies suffer from severe accuracy degradation, especially when applied to off-the-shelf ReLU-based architectures rather than models explicitly trained with QCFS Bu et al. (2022) or its distribution-aware variants (DA-QCFS) Hao et al. (2025). The core issue is a *distributional mismatch*: ANN activations are continuous-valued and static, while SNN representations are discrete and rely on temporally averaged firing rates. Conversion assumes that firing rates faithfully approximate ANN activations, but this assumption breaks down under thresholding and discrete dynamics. The resulting approximation errors accumulate across layers, severely reducing accuracy.

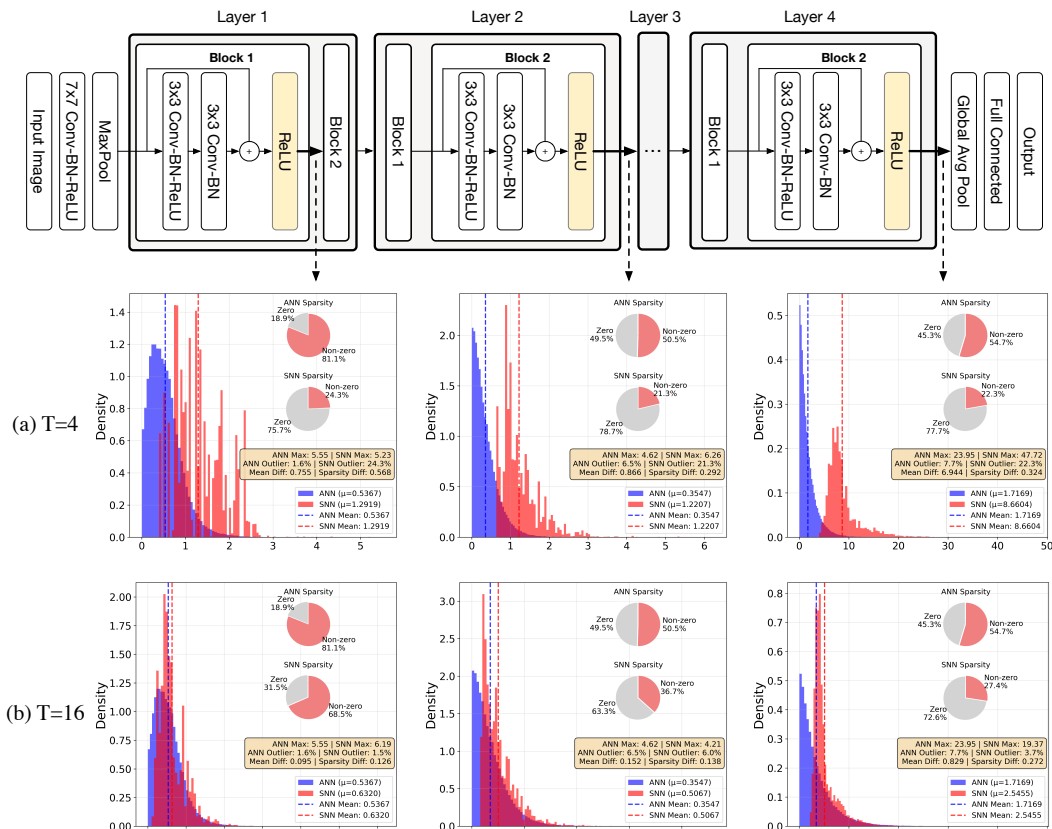

Figure 1: **Distributional Mismatch Between ANN Activations and SNN Firing Rates in ResNet-18 Under *Parallel Conversion*.** We compare the output distributions of three layers of the network — an early (2nd), middle (8th), and final (16th) layer—from a pre-trained ReLU-based ANN and their corresponding *Parallel IF* layers in the converted *parallel SNNs* (Hao et al., 2025). **(a)** The blue histograms show the distribution of the original ANN's ReLU activation values. **(b)** The red histograms show the distribution of the parallel SNN's average firing rate. The analysis is conducted at ultra-low timesteps of $T = 4$ (top) and $T = 16$ (bottom). At $T = 4$, the SNN firing rates are compressed and misaligned with the ANN's, and activation sparsity increases dramatically (e.g., from 18.9% to 75.7% in the 2nd ReLU/*Parallel IF*). Increasing the timestep to $T = 16$ partially mitigates this shift but fails to fully restore the original distribution, highlighting a fundamental challenge in low-latency conversion that our work addresses.

We illustrate this phenomenon in Figure 1, which compares activations from a pre-trained ResNet-18 and its converted parallel SNN counterpart. At $T = 4$, the SNN firing-rate distributions are compressed, shifted, and markedly sparser, indicating a loss of representational richness. Increasing $T$ to 16 partially alleviates the mismatch but undermines the benefit of low latency. Thus, the coupled *spatiotemporal distributional shift* between ANN activations and SNN responses emerges as the central obstacle to accurate, low-latency inference.

Although both classical serial ANN-to-SNN conversion methods and the first generation of parallel conversion frameworks have started to address spatial and temporal mismatches, their treatments remain largely isolated and typically yield substantial gains only when the ANN backbone has been retrained with QCFS-like or distribution-aware (DA-QCFS) activations Hao et al. (2025), while *off-the-shelf* ReLU-based architectures still suffer from poor accuracy at ultra-low timesteps.

In contrast, we explicitly target their interplay with a two-stage correction framework. First, **Spatial Recalibration** replaces ReLUs in an *off-the-shelf* ANN with DA-QCFS activations and uses activations from the parallel SNN to recalibrate the associated batch normalization statistics and affine parameters, stabilizing spike-domain feature distributions across layers. Second, **Temporal Correction** learns a lightweight per-channel bias over the time-collapsed membrane potential, injected

immediately before spike generation; trained with surrogate gradients, this bias shifts potential trajectories to cancel timestep-dependent errors introduced by the one-shot firing rule, while preserving the constant-time nature of parallel inference.

Our contributions are threefold:

1. We identify and empirically characterize the spatiotemporal distributional shift that underlies the performance gap in *parallel ANN-to-SNN conversion*.

2. We propose a two-stage correction framework—spatial recalibration of batch normalization statistics and temporal bias correction—that jointly resolves these discrepancies.

3. Our approach is highly efficient, requiring fine-tuning of only a small parameter subset. On ImageNet, for *off-the-shelf* ReLU-based backbones, we raise the top-1 accuracy of ResNet-18 at $T = 4$ **from 25.20% to 62.28**% and ResNet-34 at $T = 8$ **from 50.67% to 68.23%**, setting a new state of the art for ultra-low-latency parallel conversion. Moreover, when starting from DA-QCFS-trained ANN backbones, our calibration further boosts the best existing parallel baseline: at $T = 2$, ResNet-34 improves from **42.45% to 72.35**% and VGG-16 from **36.98% to 65.19**%.

## 2 RELATED WORK

Research on ANN-to-SNN conversion and low-latency spiking inference has advanced along two tightly coupled directions: (i) parallel and efficient architectures that eliminate temporal bottlenecks, and (ii) methods that mitigate spatial and temporal mismatches introduced by quantization, thresholds, and discrete dynamics. On the parallelization side, constant-time spiking computation and conversion frameworks demonstrate that inference complexity can be decoupled from the number of timesteps, enabling ultra-low-latency inference Hao et al. (2025). Complementary efforts develop training and architectural techniques tailored for parallel SNNs, including constant-time parallel training Feng et al. (2025), multi-parallel implicit stream architectures for efficient optimization Cao et al. (2024), Spiking State-Space Models for long sequence processing Shen et al. (2025), parallel spiking neurons designed to capture long-term dependencies Fang et al. (2023), and temporally reversible SNNs that trade training memory for $O(1)$ inference Hu et al. (2024). While these approaches substantially reduce simulation and training cost, they primarily address computational efficiency rather than conversion fidelity.

A complementary line of work focuses directly on improving fidelity in ANN-to-SNN conversion. Early studies identified errors from clipping, flooring, and thresholding, and introduced remedies such as threshold balancing and quantization-aware adjustments Li et al. (2021a); Bu et al. (2022). More recent methods address temporal mismatches: for example, initial membrane-potential shifts and offset-spike calibration mitigate one-spike discrepancies Hao et al. (2023a), while forward temporal bias calibration introduces timestep-wise biases to correct firing-rate drift without costly backpropagation through time (BPTT) Wu et al. (2024). Other strategies exploit distillation or attention to align intermediate ANN and SNN features Hong & Wang (2025), or adopt phase-coding and one-spike encoding schemes to minimize conversion loss Hwang & Kung (2024). Beyond conversion, *Guo et al.* introduce MPBN Guo et al. (2023)—a *direct-from-scratch* SNN training method that normalizes the pre-firing membrane potential and folds the normalization into firing thresholds for inference, unlike conversion methods that calibrate around fixed ANN features. In contrast to these efforts, our work is the first to explicitly analyze and jointly correct the spatiotemporal distributional mismatch that arises in *parallel* ANN-to-SNN conversion of *off-the-shelf* ReLU-based backbones.

## 3 PRELIMINARIES

**LIF Neuron Dynamics and Serial Computation.** We consider an $L$-layer Spiking Neural Network (SNN) composed of Leaky Integrate-and-Fire (LIF) neurons Hodgkin & Huxley (1952), evolving over $T$ discrete time steps. For a neuron in layer $l$ at time $t$, its dynamics are governed by:

$$\mathbf{I}^{l,t} = \mathbf{W}^l \mathbf{s}^{l-1,t} \theta^{l-1}, \qquad \mathbf{v}_{\text{pre}}^{l,t} = \lambda^l \mathbf{v}^{l,t-1} + \mathbf{I}^{l,t}, \tag{1a}$$

$$\mathbf{s}^{l,t} = \Theta\left(\mathbf{v}_{\text{pre}}^{l,t} - \theta^l\right), \qquad \mathbf{v}^{l,t} = \mathbf{v}_{\text{pre}}^{l,t} - \mathbf{s}^{l,t} \theta^l. \tag{1b}$$

where $\mathbf{v}^{l,t}$ is the membrane potential, $\mathbf{I}^{l,t}$ is the input current, $\mathbf{s}^{l,t}$ is the binary spike train, $\theta^l$ is the firing threshold, and $\lambda^l$ is the leakage factor. The temporal recurrence in (1) necessitates sequential computation across timesteps, making it the primary performance bottleneck for conventional SNN inference.

**Rate-Based Conversion and QCFS Activation.** ANN-to-SNN conversion methods bridge the two paradigms by equating ANN activations with SNN firing rates. A central tool for enabling low-latency, high-fidelity conversion is the Quantization–Clip–Floor–Shift (QCFS) activation Bu et al. (2022):

$$\mathbf{a}_{\mathrm{QCFS}}^l = \frac{\theta^l}{\tilde{T}} \, \mathrm{Clip}\left( \left\lfloor \frac{\mathbf{W}^l \mathbf{a}^{l-1}\tilde{T} + \boldsymbol{\psi}^l}{\theta^l} \right\rfloor, 0, \tilde{T} \right), \tag{2}$$

where $\boldsymbol{\psi}^l$ is a learnable channel-wise shift. This formulation defines an integer spike target

$$k^l = \mathrm{Clip}\left( \left\lfloor \frac{\mathbf{W}^l \mathbf{a}^{l-1}\tilde{T} + \boldsymbol{\psi}^l}{\theta^l} \right\rfloor, 0, \tilde{T} \right), \tag{3}$$

to be matched by the converted SNN.

A distribution-aware variant of QCFS (DA-QCFS) was proposed in Hao et al. (2025), which augments QCFS with channel-wise scaling and shifting to better match activation statistics:

$$\mathbf{a}_{\mathrm{DA}}^{l,\tilde{T}} = \frac{\theta^l + \phi_{\mathrm{DA}}^l}{\tilde{T}} \mathrm{Clip}\left( \left\lfloor \frac{(\mathbf{W}^l \mathbf{a}^{(l-1),\tilde{T}} + \psi_{\mathrm{DA}}^l)\tilde{T} + \boldsymbol{\psi}^l}{\theta^l} \right\rfloor, 0, \tilde{T} \right). \tag{4}$$

Here $\phi_{\mathrm{DA}}^l, \psi_{\mathrm{DA}}^l \in \mathbb{R}^C$ denote learnable per-channel scaling and shifting factors ($C$ is the number of channels). They are iteratively optimized using mean conversion errors before and after activation, aiming to align SNN firing-rate distributions with those of the pretrained ANN. Nonetheless, such distribution reshaping remains *suboptimal*, as it does not fully address the structural mismatch between ANN activations and SNN firing statistics.

*Assumptions underpinning QCFS.* Two premises are standard when analyzing the "optimal" shift and the zero-mean conversion error in (2) and (3). *Parameter matching:* the same threshold parameter $\theta^l$ is used both to scale ANN activations in (2) and as the firing threshold in the SNN dynamics of (1b). In addition, the membrane potential is initialized at the bin midpoint, $\mathbf{v}^{l,0} = \frac{1}{2}\theta^l$, so that the shift term $\boldsymbol{\psi}^l$ corresponds to mid-bin rounding. *Uniform in-bin statistics:* the pre-activation $\mathbf{W}^l \mathbf{a}^{l-1}$ (or its integer quantization) is assumed to be uniformly distributed within each quantization interval $\left[(m-1)\frac{\theta^l}{\tilde{T}}, m\frac{\theta^l}{\tilde{T}}\right]$, for $m = 1, \dots, \tilde{T}$. Under these premises, setting $\boldsymbol{\psi}^l$ to the mid-level value $\frac{1}{2}\theta^l$ makes rounding errors cancel in expectation, yielding zero expected conversion error for arbitrary $T$ and $\tilde{T}$, and exact equality when $T = \tilde{T}$.

**The Parallel Conversion Framework and its Lossless Mapping.** Recently, the QCFS conversion framework was extended for parallel inference (Hao et al., 2025), i.e., enabling the computation of spike targets in *one-shot* rather than sequentially. Concretely, the entire output spike train $\mathbf{s}^l = [\mathbf{s}^{l,1}, \dots, \mathbf{s}^{l,T}]^\top$ is generated through a single thresholding operation:

$$\mathbf{s}^l = \Theta\left( \boldsymbol{\Lambda}_{\mathrm{pc}}^l \mathbf{I}^l + \mathbf{b}^l - \theta^l \right), \tag{5}$$

where $\mathbf{I}^l$ is the temporal sequence of input currents and

$$\boldsymbol{\Lambda}_{\mathrm{pc}}^l = \begin{bmatrix} \frac{1}{T} & \cdots & \frac{1}{T} \\ \frac{1}{T-1} & \cdots & \frac{1}{T-1} \\ \vdots & \ddots & \vdots \\ 1 & \cdots & 1 \end{bmatrix} \tag{6}$$

$$\mathbf{b}^l = \underbrace{\left[ \frac{\boldsymbol{\psi}^l}{T}, \frac{\boldsymbol{\psi}^l}{T-1}, \dots, \boldsymbol{\psi}^l \right]^\top}_{\text{Our Motivation: QCFS-derived time-dependent bias (fixed, suboptimal)}}. \tag{7}$$

A neuron fires at time $t$ iff

$$[\mathbf{\Lambda}_{\mathrm{pc}}^l \mathbf{I}^l]_t + b^{l,t} \geq \theta^l \iff \frac{1}{T-t+1}\left(\sum_{j=1}^{T} \mathbf{I}^{l,j} + \boldsymbol{\psi}^l\right) \geq \theta^l \iff \sum_{j=1}^{T} \mathbf{I}^{l,j} + \boldsymbol{\psi}^l \geq (T-t+1)\theta^l,$$

(8)

and, using $\sum_j \mathbf{I}^{l,j} \approx \mathbf{W}^l \mathbf{a}^{l-1} T$ with $T = \tilde{T}$ for simplicity, the total spike count equals the QCFS target:

$$\sum_{t=1}^{T} s^{l,t} = \mathrm{Clip}\left(\left\lfloor \frac{\mathbf{W}^l \mathbf{a}^{l-1} T + \boldsymbol{\psi}^l}{\theta^l} \right\rfloor, 0, T\right) = k^l.$$

(9)

Thus, the *lossless* mapping of the parallel conversion is anchored in the same target construction as QCFS. In particular, analyses of its lossless and *sorting* properties, as well as derivations of the *optimal shifting distance* per step, implicitly depend on QCFS premises: *parameter matching* between activation range and threshold, and *uniform in-bin* statistics for rounding. As we show next, these assumptions are fragile in practice, especially at ultra-low $T$—and the aggregation weights in (5) make the precomputed *time-invariant* shift $\psi^l$ suboptimal. This motivates our distribution-aware calibration in Sec. 4, which corrects the resulting step-dependent bias without altering the constant-time mapping.

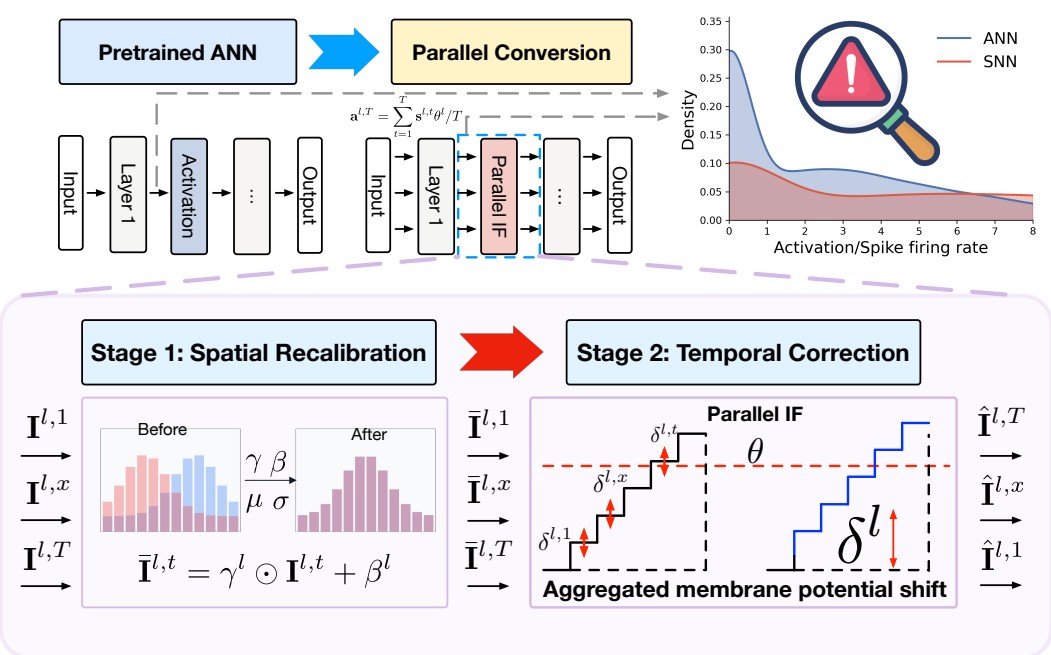

Figure 2: **Overall pipeline motivated by the *challenges* in parallel ANN-to-SNN conversion. Left (Observed issues).** Converting a pretrained ANN to a *parallel* SNN replaces serial accumulation with a one–shot test against a *descending threshold ladder* $(T-t+1)\theta$, which compresses activation ranges and amplifies late–timestep sensitivity. This creates two key discrepancies: (i) a *spatial* shift, as normalization layers expect ANN statistics but encounter spike–domain ones; and (ii) a *temporal* bias, as rounding residuals are distributed unevenly across timesteps. **Right (Remedy).** The pipeline addresses these causes by aligning operator–induced feature moments with backbone expectations and compensating ladder–induced bias in a channel–wise, time–collapsed form. By correcting the *why*—statistical shift and threshold asymmetry—it restores consistency at ultra–low $T$ while preserving constant–time, one–shot firing.

## 4 METHOD

### 4.1 OVERVIEW

Our goal is to improve the fidelity of *parallel* ANN-to-SNN conversion at ultra-low timesteps $T$, especially when starting from *off-the-shelf* ReLU-based backbones. We consider two types of pretrained backbones: (i) *off-the-shelf* ReLU-based ANNs and (ii) ANNs retrained with QCFS activations and subsequently calibrated with DA-QCFS.

• **Stage 1: Spatial recalibration.** In both settings, we bring the backbone into a DA-QCFS form and recalibrate the BatchNorm running statistics and affine parameters before DA-QCFS activation layers, thereby aligning the normalized activation statistics with the spike-domain firing-rate targets used in subsequent parallel conversion.
• **Stage 2: Temporal correction.** The parallel firing rule in Eq. 5 already induces a time-dependent bias determined by the IF dynamics; we augment this formulation with a lightweight per-channel trainable membrane-potential bias, applied immediately before spike generation, to correct the timestep-dependent skew characterized in Eq. 7.

The detailed derivations that connect QCFS, serial IF dynamics and the parallel neuron coefficients are provided in Appendix A; here we focus on the concrete operations applied in our method.

### 4.2 STAGE 1: BATCHNORM (BN) RECALIBRATION WITH DA-QCFS

**BN mismatch under spike-domain activations.** Given a pretrained ANN backbone (either *off-the-shelf* ReLU-based or QCFS/DA-QCFS-based as described above), we convert it to DA-QCFS-based ANN. But BN layers still use running statistics $(\mu^l, \sigma^l)$ estimated from the original pretrained ANN activations. At small $T$ after parallel SNN conversion, spike-domain activations and their empirical mean and variance can deviate significantly from the ANN regime. This means that the same $(\mu^l, \sigma^l)$ no longer normalize the features properly, introducing a systematic shift that accumulates with depth.

**Recalibrating BN statistics in the DA-QCFS ANN.** To address this issue, we recalibrate BN statistics in the DA-QCFS ANN that serves as the rate proxy for the subsequent parallel SNN. Concretely, we switch all BN layers to training mode, freeze all non-BN weights, and run a calibration set $\mathcal{D}_{cal}$ through the DA-QCFS ANN. Let $\Phi_{DA}^l(x)$ denote the BN input at layer $l$. We estimate recalibrated statistics as

$$\widehat{\mu}_s^l = \frac{1}{|\mathcal{D}_{cal}|} \sum_{x \in \mathcal{D}_{cal}} \Phi_{DA}^l(x), \qquad (\widehat{\sigma}_s^l)^2 = \frac{1}{|\mathcal{D}_{cal}|} \sum_{x \in \mathcal{D}_{cal}} \left(\Phi_{DA}^l(x) - \widehat{\mu}_s^l\right)^2, \qquad (10)$$

and overwrite the running statistics, $(\mu^l, \sigma^l) \leftarrow (\widehat{\mu}_s^l, \widehat{\sigma}_s^l)$. We then fine-tune only the BN parameters $(\mu^l, \sigma^l, \gamma^l, \beta^l)$ for a few epochs on $\mathcal{D}_{cal}$ under the same DA-QCFS ANN forward, keeping all convolutional and linear weights fixed. Statistics are re-estimated and BN affine parameters are adapted under the same DA-QCFS ANN that will be used at inference. Empirically, this removes most of the spatial distortion at low $T$ and recovers stable feature distributions across depth.

### 4.3 STAGE 2: TEMPORAL BIAS CORRECTION FOR PARALLEL FIRING

**Analytical form of the parallel neuron.** The parallel firing rule in Eq. (5) can be understood directly from the serial IF dynamics. Starting from the discrete-time update

$$v^t = v^{t-1} + I - s^t\theta, \qquad (11)$$

unrolling over $T$ steps gives (see Eq. (20))

$$v^T = v^0 + T \cdot I - \left(\sum_{k=1}^{T} s^k\right)\theta. \qquad (12)$$

If the neuron fires for the first time at step $t$, then it will fire at all subsequent steps, and the total spike count is $N = T - t + 1$. Requiring $v^T \geq 0$ at that point yields the firing condition

$$v^0 + T \cdot I \geq (T - t + 1)\theta. \qquad (13)$$

Normalizing by $(T - t + 1)$ leads to a standard neuron form

$$\frac{T}{T - t + 1}I + \frac{v^0}{T - t + 1} \geq \theta. \tag{14}$$

More generally, when the input current varies over time, we can replace $TI$ by the cumulative current $U^l = \sum_{j=1}^{T} I^{l,j}$ (or, equivalently, view $I$ as the time-averaged current). Identifying the QCFS shift $v^0 = \psi$ and using $U^l$, we rewrite the pre-threshold potential in layer $l$ at step $t$ as

$$v_{\text{pre}}^{l,t} = \frac{U^l + \psi^l}{T - t + 1}, \qquad s^{l,t} = \Theta\big(v_{\text{pre}}^{l,t} - \theta^l\big). \tag{15}$$

which is exactly the parallel rule in Eq. (5) written as a neuron with *time-dependent* weight and bias

$$\Lambda_{\text{PC}}(t) = \frac{T}{T - t + 1}, \qquad b^l(t) = \frac{\psi^l}{T - t + 1}. \tag{16}$$

These coefficients are not arbitrary quantization parameters: they are analytically determined by the IF dynamics and the QCFS form. Appendix A gives a detailed step-by-step derivation and a side-by-side comparison with generic activation quantization.

**Timestep-dependent skew and our correction.** Because both the effective weight and bias depend on $(T - t + 1)$, the decision boundary changes over time: early timesteps use a high effective threshold, while late timesteps use a much lower one. As we show in Appendix A, this induces a systematic temporal skew: spikes tend to be delayed and concentrate in the last $N$ slots, even when the total spike count matches the QCFS target. At very small $T$, this skew causes a noticeable mismatch between the temporal firing pattern of the parallel SNN and the rate implied by the DA-QCFS activation.

To compensate for this skew without changing the constant-time mapping, we keep the analytically derived aggregation matrix $\boldsymbol{\Lambda}_{\text{pc}}^l$ fixed and modify only the bias term. For each layer $l$ and channel $c \in \{1, \ldots, C_l\}$, we introduce a light-weight output-side correction $\Delta b_c^l$ and redefine the time-dependent bias as

$$\tilde{b}_c^l(t) = \frac{\psi^l}{T - t + 1} + \Delta b_c^l, \qquad t = 1, \ldots, T, \tag{17}$$

or, in vector form,

$$\tilde{\mathbf{b}}^l(t) = \mathbf{b}^l(t) + \Delta \mathbf{b}^l, \qquad \Delta \mathbf{b}^l \in \mathbb{R}^{C_l}. \tag{18}$$

Intuitively, $\mathbf{b}^l(t)$ encodes the ideal time-varying bias implied by the IF dynamics and QCFS, while $\Delta \mathbf{b}^l$ is a learnable per-channel correction that shifts the entire ladder for that channel.

**Training and complexity.** We optimize $\Delta \mathbf{b}^l$ using standard surrogate gradients, while freezing all other parameters in the backbone. In practice we use the same surrogate function as in our SNN baselines and backpropagate the task loss through the parallel firing rule with $\tilde{\mathbf{b}}^l(t)$. This adds only $O(C_l)$ parameters per layer (independent of $T$) and does not change the constant-time inference cost: at test time, the only difference from the original parallel rule is the addition of $\Delta \mathbf{b}^l$ to the bias vector before thresholding.

Together, Stage 1 and Stage 2 form a distribution-aware calibration procedure: Stage 1 aligns spatial statistics of BN, and Stage 2 corrects the timestep-dependent bias, without modifying the aggregation matrix or the one-shot firing rule.

## 5 EXPERIMENT

We validate the performance of our proposed method on the CIFAR-10/100 Krizhevsky et al. (2009) and ImageNet Deng et al. (2009) datasets, utilizing common VGG Simonyan & Zisserman (2014) and ResNet He et al. (2016) architectures. Our approach is benchmarked against a comprehensive set of state-of-the-art SNN training and conversion paradigms. These include direct training methods (e.g., TAB Jiang et al. (2024), TTS Guo et al. (2024)), hybrid training (e.g., LM-H Hao et al. (2023b)), conversion with subsequent rectification (e.g., SNM Wang et al. (2022), FTBC Wu et al. (2024)), and both standard (e.g., QCFS Bu et al. (2022), TPP Bojkovic et al. (2025)) and parallel ANN-to-SNN conversion (FS-PC Hao et al. (2025)). Detailed experimental configurations and hyperparameters are provided in the Appendix.

Table 1: Comparison of our method with other state-of-the-art approaches on CIFAR-10, CIFAR-100, and ImageNet. The symbol † indicates methods that adopt the error calibration defined in Eq. 4 from Hao et al. (2025).

| Dataset | Method | Type | ANN Acc.(%) | Arch. | T | SNN Acc.(%) |
|---|---|---|---|---|---|---|
| CIFAR-10 | QCFS Bu et al. (2022) [ICLR] | ANN-SNN Conversion | 95.52 | VGG-16 | 2, 4, 8 | 91.18, 93.96, 94.95 |
| | FTBC Wu et al. (2024) [ECCV] | Conversion Rect. | 95.92 | VGG-16 | 2, 4 | 92.08, 94.67 |
| | SNM Wang et al. (2022) [ICLR] | Conversion Rect. | 94.09 | VGG-16 | 32 | 93.43 |
| | FS-PC Hao et al. (2025) [ICML] | Parallel Conversion | 95.43 | VGG-16 | 2 | 94.16 |
| | **Ours** | **Parallel Conversion** | 95.43 | VGG-16 | 2 | **94.32** |
| CIFAR-100 | LM-H Hao et al. (2023b) [ICLR] | Hybrid Training | - | VGG-16 | 4 | 73.11 |
| | QCFS Bu et al. (2022) [ICLR] | ANN-SNN Conversion | 76.28 | VGG-16 | 2, 4, 8 | 63.79, 69.62, 73.96 |
| | TPP Bojkovic et al. (2025) [ICML] | ANN-SNN Conversion | 76.21 | VGG-16 | 4, 8 | 73.93, 76.03 |
| | SNM Wang et al. (2022) [ICLR] | Conversion Rect. | 74.13 | VGG-16 | 32 | 71.80 |
| | FTBC Wu et al. (2024) [ECCV] | Conversion Rect. | 76.21 | VGG-16 | 4, 8 | 71.47, 75.12 |
| | FS-PC Hao et al. (2025) [ICML] | Parallel Conversion | 76.11 | VGG-16 | 2, 4 | 72.71, 75.98 |
| | **Ours** | **Parallel Conversion** | 76.11 | VGG-16 | 2, 4 | **74.03, 76.42** |
| ImageNet-1k | QCFS Bu et al. (2022) [ICLR] | ANN-SNN Conversion | 74.29 | VGG-16 | 8, 16, 32 | 19.12, 50.97, 68.47 |
| | SNM Wang et al. (2022) [ICLR] | Conversion Rect. | 73.18 | VGG-16 | 32 | 64.78 |
| | Burst Li & Zeng (2022) [IJCAI] | Conversion Rect. | 74.27 | VGG-16 | 32 | 70.61 |
| | FTBC Wu et al. (2024) [ECCV] | Conversion Rect. | 73.91 | VGG-16 | 8, 16 | 69.31, 72.98 |
| | FS-PC Hao et al. (2025) [ICML] | Parallel Conversion | 74.23 | VGG-16 | 2, 4 | 36.93, 71.23 |
| | FS-PC† Hao et al. (2025) [ICML] | Parallel Conversion | 74.23 | VGG-16 | 2, 4 | 56.50, 71.75 |
| | **Ours** | **Parallel Conversion** | 74.23 | VGG-16 | 2, 4 | **63.91, 72.23** |
| | **Ours†** | **Parallel Conversion** | 74.23 | VGG-16 | 2, 4 | **61.84, 72.65** |
| | Dspike Li et al. (2021b) [NeurIPS] | Direct Training | - | ResNet-34 | 6 | 68.19 |
| | RecDis Guo et al. (2022) [CVPR] | Direct Training | - | ResNet-34 | 6 | 67.33 |
| | GLIF Yao et al. (2022) [NeurIPS] | Direct Training | - | ResNet-34 | 4 | 67.52 |
| | TAB Jiang et al. (2024) [ICLR] | Direct Training | - | ResNet-34 | 4 | 67.78 |
| | SEENN-I Li et al. (2023) [NeurIPS] | Direct Training | - | ResNet-34 | 3.38 | 64.66 |
| | GAC-SNN Qiu et al. (2024) [AAAI] | Direct Training | - | ResNet-34 | 6 | 70.42 |
| | TTS Guo et al. (2024) [AAAI] | Direct Training | - | ResNet-34 | 4 | 70.74 |
| | QCFS Bu et al. (2022) [ICLR] | ANN-SNN Conversion | 74.32 | ResNet-34 | 8, 16, 32 | 35.06, 59.35, 69.37 |
| | FTBC Wu et al. (2024) [ECCV] | Conversion Rect. | 74.32 | ResNet-34 | 8, 16 | 65.28, 71.66 |
| | TPP Bojkovic et al. (2025) [ICML] | ANN-SNN Conversion | 74.32 | ResNet-34 | 8, 16 | 67.32, 72.03 |
| | FS-PC Hao et al. (2025) [ICML] | Parallel Conversion | 74.30 | ResNet-34 | 2, 4 | 42.45, 67.28 |
| | FS-PC† Hao et al. (2025) [ICML] | Parallel Conversion | 74.30 | ResNet-34 | 2, 4 | 65.20, 72.90 |
| | **Ours** | **Parallel Conversion** | 74.30 | ResNet-34 | 2, 4 | **68.41, 73.24** |
| | **Ours†** | **Parallel Conversion** | 74.30 | ResNet-34 | 2, 4 | **69.27, 73.10** |

## 5.1 COMPARISON WITH STATE-OF-THE-ART METHODS

We first evaluate our method using pre-trained ANNs with the QCFS activation function, a common practice in recent conversion works designed to achieve high performance at low latency. The results are presented in Table 1.

**CIFAR Datasets:** On CIFAR-100 with a VGG-16 backbone, our method achieves 74.03% ($T = 2$) and 76.42% ($T = 4$) accuracy. This surpasses the prior parallel conversion baseline (FS-PC) and outperforms strong conversion rectification (FTBC) and direct conversion (TPP) methods at identical timesteps. We observe similar state-of-the-art performance on CIFAR-10.

**ImageNet Dataset:** The advantages of our method are most prominent on ImageNet. For ResNet-34 at $T = 4$, we achieve 73.24% accuracy, exceeding leading direct training methods like TTS (70.74%) and GAC-SNN (70.42%) without requiring complex temporal-domain optimization. With calibration (†), our model reaches 73.10% at $T = 4$, consistently outperforming the FS-PC baseline. Substantial gains at ultra-low latencies are also observed on the VGG-16 architecture.

## 5.2 PERFORMANCE ON STANDARD MODELS

To demonstrate the generalizability of our method beyond specialized activation functions, we evaluate its performance on standard ANNs pre-trained with the conventional ReLU activation. This is a *more challenging* and practical scenario due to the larger potential for quantization errors during conversion. As shown in Table 2, our method consistently and significantly outperforms the FS-PC baseline across various ResNet architectures. For instance, on ResNet-34 at $T = 8$, our method reduces the accuracy drop from the source ANN from 22.64% down to 5.07%. This highlights our method's robustness and its ability to mitigate conversion errors effectively, even without modifications to the source ANN's architecture or activation functions.

Table 2: Comparison of conversion algorithms on ReLU-based ResNet models (ImageNet-1k).

| Arch. | ANN Acc.(%) | FS-PC Hao et al. (2025) | | Ours | |
|---|---|---|---|---|---|
| | | $T = 8$ | $T = 16$ | $T = 8$ | $T = 16$ |
| ResNet-18 | 69.76 | 55.18 (-14.58) | 66.26 (-3.50) | **68.25** (-1.51) | **68.27** (-1.49) |
| ResNet-34 | 73.31 | 50.67 (-22.64) | 68.04 (-5.27) | **68.23** (-5.07) | **71.66** (-1.64) |
| ResNet-50 | 76.12 | 64.16 (-11.96) | 73.59 (-2.53) | **70.79** (-5.36) | **73.82** (-2.33) |
| ResNet-101 | 77.38 | 60.59 (-16.79) | 73.86 (-3.52) | **67.33** (-10.04) | **73.96** (-3.41) |

Table 3: Ablation study on ReLU-based ResNet models. We report SNN accuracy for the baseline (FS-PC) and after sequentially adding our spatial recalibration, temporal correction, and both components (Full). *Additional layer-wise visualizations (2nd, 8th, and 16th layers) are provided in Appendix B.*

| Arch. | ANN Acc.(%) | $T$ | Baseline(FS-PC) | Spatial | Temporal | Full |
|---|---|---|---|---|---|---|
| ResNet-18 | 69.76 | 4 | 25.20 | 61.83 (+36.63) | 55.55 (+30.35) | 62.28 (+37.08) |
| ResNet-18 | 69.76 | 8 | 55.18 | 68.08 (+12.90) | 63.66 (+8.48) | 68.25 (+13.07) |
| ResNet-34 | 73.31 | 8 | 50.67 | 68.08 (+17.41) | 56.14 (+5.47) | 68.23 (+17.56) |
| ResNet-50 | 76.12 | 8 | 64.42 | 71.28 (+6.86) | 70.60 (+6.18) | 70.79 (+6.37) |

## 5.3 ABLATION STUDY

To dissect the individual contributions of our proposed components—*spatial recalibration* and *temporal correction*—we conduct a detailed ablation study. We evaluate four configurations on ReLU-based ResNet models: (a) a baseline parallel conversion (FS-PC), (b) the baseline with only spatial recalibration, (c) the baseline with only temporal correction, and (d) full method combining both.

The results, summarized in Table 3, reveal that both components provide substantial and complementary improvements over the baseline. On ResNet-18 at $T = 8$, spatial recalibration alone boosts accuracy from 55.18% to 68.08%, while temporal correction increases it to 63.66%. Combining both yields the best performance of 68.25%, confirming their synergistic effect. Figure 3 provides a visual analysis, illustrating how spatial recalibration aligns the magnitude of the SNN's average firing rate with the ANN's activation value, while temporal correction refines the underlying spike timing patterns. Together, they enable a more faithful emulation of the original ANN's activations, leading to higher accuracy in the resulting SNN.

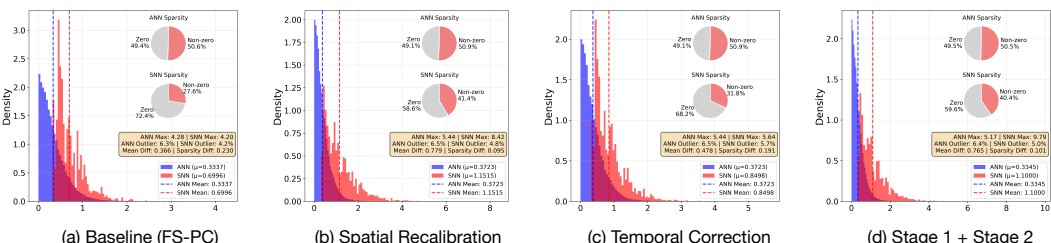

(a) Baseline (FS-PC)     (b) Spatial Recalibration     (c) Temporal Correction     (d) Stage 1 + Stage 2

Figure 3: Visualization of the proposed two-stage calibration on the 8th ReLU / parallel IF layer of a ReLU-based ResNet-18 at $T = 8$.

## 6 CONCLUSION

This paper tackles the severe accuracy degradation in parallel ANN-to-SNN conversion at ultra-low timesteps ($T \leq 8$). We trace this issue to a spatiotemporal distributional shift, where the parallel firing mechanism induces both spatial mismatches in feature statistics and a systematic temporal bias. To resolve this, we introduced a two-stage calibration that recalibrates normalization layers for the spike domain and learns a temporal correction to offset firing-time errors, all while preserving constant-time inference. Our method dramatically boosts performance, raising ResNet-18 accuracy on ImageNet from **25.20**% to **62.28**% at $T = 4$. This work demonstrates that correcting the flawed statistical assumptions of the parallel framework is key to unlocking its potential for high-fidelity, low-latency inference.

## REPRODUCIBILITY STATEMENT

We have made extensive efforts to ensure the reproducibility of our results. The main paper provides detailed descriptions of the proposed method and experiments. Additional implementation details, training configurations, and hyperparameters are documented in the appendix. Complete proofs of theoretical claims are also provided in the appendix. To facilitate replication, we will release all source code, scripts and logs in our experiments.

## ETHICS STATEMENT

This work complies with the ethical standards set forth by the ICLR community. Our research does not involve human subjects, personally identifiable information, or sensitive data. All datasets used in our experiments are publicly available and widely adopted within the community (e.g., CIFAR, ImageNet), with appropriate licenses respected. We have taken care to report results transparently, ensure reproducibility through code and documentation, and minimize potential misuse by clarifying the intended scientific purpose. We do not anticipate direct societal risks beyond those commonly associated with advances in machine learning research.

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

## USE OF LARGE LANGUAGE MODELS (LLMs)

In preparing this work, we employed large language models (LLMs) as general-purpose assistive tools. Specifically, LLMs were used for: (i) grammar polishing and stylistic refinement of the manuscript text, (ii) assistance in reviewing and debugging baseline code implementations, and (iii) occasional support in summarizing related work for clarity. Importantly, no part of the research idea, methodology design, or experimental results relied on LLMs. The conceptual contributions, technical development, and validation of this work were conducted entirely by the authors. We take full responsibility for all content presented in this paper.

## APPENDIX

## A    CLARIFICATION ON PARALLEL SNN VS. GENERIC QUANTIZATION

Ultimately, isn't this merely multiplying the input by a coefficient and comparing it against a threshold? Is this not simply quantization?

To dispel this preconception, we demonstrate how the **coefficients and bias terms of Parallel SNN are derived**, rather than being manually configured statistical parameters as in Generic Quantization.

### A.1    SERIAL IF/LIF SNN

At layer $l$ and time step $t$, the serial IF/LIF dynamics are

$$v_{\text{PRE}}^{l,t} = \lambda^l v^{l,t-1} + I^{l,t},$$
$$v^{l,t} = v_{\text{PRE}}^{l,t} - s^{l,t}\theta^l,$$
$$I^{l,t} = W^l s^{l-1,t}\theta^{l-1},$$
$$s^{l,t} = \begin{cases} 1, & v_{\text{PRE}}^{l,t} \geq \theta^l, \\ 0, & \text{otherwise.} \end{cases}$$

Over $T$ time steps, the *rate* (or spike-count normalized) of this layer is

$$r_{\text{SNN}}^{l,T} \triangleq \frac{1}{T}\sum_{t=1}^{T} s^{l,t}\theta^l.$$

### A.2    QCFS ACTIVATION (ANN SIDE)

Let $\tilde{T}$ denote the simulation time period in the QCFS function and $\psi^l$ the shift term. For simplicity, define the ANN pre-activation at layer $l$ as

$$u^{l,\tilde{T}} \triangleq W^l r^{l-1,\tilde{T}} \in \mathbb{R}^{d_l}.$$

Then Eq. (5) in the paper can be written as

$$r_{\text{QCFS}}^{l,\tilde{T}} = \frac{\theta^l}{\tilde{T}}\,\text{Clip}\left(\left\lfloor \frac{u^{l,\tilde{T}}\tilde{T} + \psi^l}{\theta^l} \right\rfloor,\ 0,\ \tilde{T}\right) \tag{5}$$

Equivalently, if we define the integer spike-count

$$q^l(u^{l,\tilde{T}}) \triangleq \text{Clip}\left(\left\lfloor \frac{u^{l,\tilde{T}}\tilde{T} + \psi^l}{\theta^l} \right\rfloor,\ 0,\ \tilde{T}\right) \in \{0,\ldots,\tilde{T}\},$$

then

$$r_{\text{QCFS}}^{l,\tilde{T}} = \frac{\theta^l}{\tilde{T}}\,q^l(u^{l,\tilde{T}}).$$

Here $q^l(u^{l,\tilde{T}})$ is an integer in $\{0,\ldots,\tilde{T}\}$ and can be interpreted as the **expected spike count**; the output $r_{\text{QCFS}}^{l,\tilde{T}}$ can only take finitely many values in $\{0, \frac{\theta^l}{\tilde{T}}, \ldots, \theta^l\}$, which is essentially a **uniform quantization** of $u^{l,\tilde{T}}$.

## A.3 PARALLEL SNN (PARALLEL NEURON)

In the parallel SNN, for each layer $l$ we design a $T$-step parallel neuron:

- First, QCFS produces the rate

$$r_{\text{QCFS}}^{l,\tilde{T}} \in \left\{0, \frac{\theta^l}{\tilde{T}}, \ldots, \theta^l\right\};$$

- Then the spike pattern of this layer over $T$ time steps is generated in one shot.

We abstract this as

$$\mathbf{s}_{\text{para}}^l = \left[s_{\text{para}}^{l,1}, \ldots, s_{\text{para}}^{l,T}\right]^\top = H\left(\Lambda_{\text{PC}}^l\, r_{\text{QCFS}}^{l,\tilde{T}} + \mathbf{b}^l - \theta^l \mathbf{1}\right)$$

- $H(\cdot)$ is the element-wise Heaviside function (it outputs 1 if the argument is $\geq 0$, and 0 otherwise);
- $\Lambda_{\text{PC}}^l \in \mathbb{R}^{T \times 1}$ is a **time-dependent coefficient matrix** (the $t$-th row corresponds to a factor such as $T/t$ or $T/(T-t+1)$);
- $\mathbf{b}^l \in \mathbb{R}^T$ is the **time-dependent shift vector** for the $l$-th layer:

$$\mathbf{b}^l = \left[\psi^l/T, \ldots, \psi^l/(T-x+1), \ldots, \psi^l\right]^\top;$$

- After obtaining the parallel spike sequence, the parallel rate of this layer is

$$r_{\text{para}}^{l,T} \triangleq \frac{1}{T}\sum_{t=1}^{T} s_{\text{para}}^{l,t}\theta^l.$$

- If $T = \tilde{T}$, then

$$r_{\text{para}}^{l,T} = r_{\text{QCFS}}^{l,\tilde{T}};$$

- If $T \neq \tilde{T}$ and $\psi^l = \theta^l/2$, then

$$\mathbb{E}\left(r_{\text{para}}^{l,T} - r_{\text{QCFS}}^{l,\tilde{T}}\right) = 0,$$

i.e., $r_{\text{para}}^{l,T}$ and $r_{\text{QCFS}}^{l,\tilde{T}}$ remain equivalent from the perspective of mathematical expectation, regardless of whether $\tilde{T}$ equals $T$ or not.

$\Lambda_{\text{PC}}^l$ and $\mathbf{b}^l$ are derived from the IF/LIF dynamics and the QCFS form, so that the parallel SNN is equivalent (or equivalent in expectation) to the serial SNN in terms of rate.

## A.4 TIME-DEPENDENT BIAS IN SNN

By starting from the dynamic equations of the IF neuron and deriving step-by-step, we prove that the parameters $\Lambda_{PC}$ and $\mathbf{b}$ of Parallel SNN necessarily carry information regarding the time dimension $T$, a characteristic completely absent in Generic Quantization.

**Starting Point: The Dynamic Difference Equation of the Serial IF Neuron**

Everything begins with the standard discrete-time difference equation of the Integrate-and-Fire (IF) neuron. Let $v^t$ be the membrane potential at time $t$, $I$ be the constant input current, $s^t \in \{0,1\}$ be the spike, and $v^0$ be the initial potential (i.e., Shift term $\psi$).

$$v^t = v^{t-1} + I - s^t\theta, \quad \text{if } v^{t-1} + I \geq \theta \text{ then } s^t = 1 \text{ else } 0 \tag{19}$$

**Unfolding the Time Axis and Finding the Accumulation Relationship**

- Time-step $t = 1$:

$$v^1 = v^0 + I - s^1\theta$$

- Time-step $t = 2$:

$$v^2 = v^1 + I - s^2\theta$$
$$\Rightarrow (v^0 + I - s^1\theta) + I - s^2\theta$$
$$\Rightarrow v^0 + 2I - (s^1 + s^2)\theta$$

- Time-step $t = 3$:

$$v^3 = v^2 + I - s^3\theta$$
$$\Rightarrow (v^0 + 2I - (s^1 + s^2)\theta) + I - s^3\theta$$
$$\Rightarrow v^0 + 3I - (s^1 + s^2 + s^3)\theta$$

$\Rightarrow$ Induction of the General Formula for Step $T$: After $T$ timesteps, the state of the membrane potential must satisfy:

$$v^T = v^0 + T \cdot I - \left(\sum_{k=1}^{T} s^k\right)\theta \tag{20}$$

**Introducing the Sorting Property and Establishing the Mapping Between Spike Count and Time**

This is the critical point where Parallel SNN diverges from Generic Quantization. The larger the input current $I$, the earlier the neuron fires. If the neuron starts firing at time $t$ ($s^t = 1$), it will fire at all subsequent times $t, t+1, \ldots, T$.

This implies:

If firing occurs for the first time at moment $t \Rightarrow$ Total spike count $N = T - t + 1$

Conversely, judging **whether it should fire at moment** $t$ is equivalent to judging **whether the total accumulated potential is sufficient to support** $T - t + 1$ **spikes**.

Substituting $N = T - t + 1$ into (Eq. equation 20), and using the firing condition $v^T \geq 0$ (i.e., potential not exhausted):

$$v^0 + T \cdot I - (T - t + 1)\theta \geq 0$$
$$\Rightarrow \quad v^0 + T \cdot I \geq (T - t + 1)\theta \tag{21}$$

**Constructing the Parallel Form and The Natural Emergence of Time-Dependent Parameters**

One might consider Eq. equation 21 to be merely an inequality. We need to transform it into the **Weight** and **Bias** form of a neuron to prove the time dependency of the parameters. We divide both sides of (Eq. equation 21) by $(T - t + 1)$ to normalize the threshold $\theta$:

$$\frac{v^0 + T \cdot I}{T - t + 1} \geq \theta$$
$$\Rightarrow \quad \left(\frac{T}{T - t + 1}\right) \cdot I + \left(\frac{v^0}{T - t + 1}\right) \geq \theta$$

Now, we compare this with the standard neuron formula $\mathbf{W}x + \mathbf{b} \geq \theta$:

1. Equivalent Weight (Time-dependent):

$$\Lambda_{PC}(t) = \frac{T}{T - t + 1}$$

- When $t = 1$, the weight is 1.
- When $t = T$, the weight is $T$.
- $\Rightarrow$ **The coefficient changes with** $t$.

2. Equivalent Bias (Time-dependent): Let $v^0 = \psi$ (QCFS shift term), then:

$$\mathbf{b}(t) = \frac{\psi}{T - t + 1}$$

- When $t = 1$, the Bias is $\psi/T$.
- When $t = T$, the Bias is $\psi$.
- $\Rightarrow$ **The Bias changes with $t$.**

In addition to the analytically derived time-dependent bias, we further introduce a *trainable* output-side correction $\Delta b_c^l$ for each channel $c$, and redefine the time-dependent bias as

$$\tilde{b}_c^l(t) = \frac{\psi^l}{T - t + 1} + \Delta b_c^l, \qquad t = 1, \dots, T.$$

Equivalently, in vector form,

$$\tilde{\mathbf{b}}^l(t) = \mathbf{b}^l(t) + \Delta \mathbf{b}^l,$$

where $\mathbf{b}^l(t)$ is the analytically derived time-dependent bias from the IF/LIF dynamics, and $\Delta \mathbf{b}^l \in \mathbb{R}^C$ is a light-weight learnable correction broadcast across the $T$ time steps.

**The Definitive Comparison and Formula Comparison with Generic Quantization**

Now we place the derivation results directly in comparison with Generic Quantization:

- Generic Quantization (Static):
$$q = \text{clip}\left(\frac{x}{s} + z\right)$$
  Here $s$ (scale) and $z$ (zero-point) are **constant scalars** for all samples and all "times" (if such a dimension exists). It has no origin in dynamics.
- Parallel SNN (Dynamic derived):
$$\mathbf{s}^t = H\left(\Lambda_{PC}(t) \cdot I + \mathbf{b}(t) - \theta\right)$$
  Here $\Lambda_{PC}(t)$ and $\mathbf{b}(t)$ are **functions of time** $t$. Their values are strictly constrained by the accumulation summation process $(T - t + 1)$ in Step 2.

In summary, Generic Quantization is a **statistical approximation in space**, whereas Parallel SNN is an **analytical expansion of temporal dynamics**. We need to consider the influence of timestep $t$ on the accumulated potential (i.e., the denominator $T - t + 1$) to derive the Parallel SNN formula. Detailed comparison is shown in Table 4 Therefore, Parallel SNN is not Generic Quantization.

| Concept / role | Generic activation quantization | Parallel SNN neuron (this work) |
|---|---|---|
| Continuous input | $x$ (float activation) | ANN pre-activation $u^{l,\tilde{T}}$ or time-averaged input current |
| Discrete / integer representation | $q \in \mathbb{Z}$ | Integer spike-count $q^l(u^{l,\tilde{T}})$ or discretized rate $r_{\text{QCFS}}^{l,\tilde{T}} = \frac{\theta^l}{\tilde{T}} q^l(u^{l,\tilde{T}})$ |
| Value range / scale | Scale $s$ (per-tensor / per-channel) | Effective step size $\theta^l/\tilde{T}$ and upper bound $\theta^l$ |
| Shift / zero-point | Scalar zero-point $z$ | Time-dependent shift vector $\mathbf{b}^l \in \mathbb{R}^T$ (e.g., $[\psi^l/T, \ldots, \psi^l]^\top$) |
| Time dimension | None (static mapping $x \to \hat{x}$) | Explicit time steps $t = 1, \ldots, T$ and coefficient matrix $\Lambda_{\text{PC}}^l(t)$ |
| Output "reconstructed" value | $\hat{x} = s(q - z)$ | Parallel rate $r_{\text{para}}^{l,T} = \frac{1}{T} \sum_{t=1}^{T} s_{\text{para}}^{l,t} \theta^l$ |
| Output 0/1 spike sequence | Not defined | $\mathbf{s}_{\text{para}}^l = H(\Lambda_{\text{PC}}^l r_{\text{QCFS}}^{l,\tilde{T}} + \mathbf{b}^l - \theta^l \mathbf{1}) \in \{0,1\}^T$ |
| Source of parameters | $(s, z)$ calibrated to minimize reconstruction error (e.g., $\mathbb{E}\|x - \hat{x}\|^2$) | $(\Lambda_{\text{PC}}^l, \mathbf{b}^l, \psi^l)$ analytically derived from IF/LIF dynamics and QCFS |
| Relation to serial SNN spike behavior | No notion of spikes or SNN equivalence | Rate-equivalence constraints $r_{\text{para}}^{l,T} \approx r_{\text{QCFS}}^{l,\tilde{T}} \approx r_{\text{SNN}}^{l,T}$ |

Table 4: Side-by-side comparison of generic activation quantization and the proposed parallel SNN neuron

## B  ABLATION STUDY

To provide a deeper insight into our ablation study, Figure 4 visualizes the impact of each proposed component on the output distributions at different depths of the network. We compare the source ANN's activation distributions (blue) with the converted SNN's average firing rates (red) at an early (2nd), middle (8th), and final (16th) layer of ResNet-18.

The baseline parallel conversion (a-c) exhibits a severe distribution mismatch, where the SNN's firing rates are significantly lower and sparser than the target ANN activations. Applying only Stage 1, Spatial Recalibration (d-f), almost completely resolves this magnitude discrepancy by aligning the means of the two distributions, demonstrating its role in correcting the output scale. While Stage 2, Temporal Correction (g-i), has a less pronounced effect on the static distribution by itself, combining it with Stage 1 (j-l) results in the most faithful emulation. Our full method consistently produces SNN firing rate distributions that closely match the ANN's statistics across all observed layers, effectively minimizing conversion error throughout the network.

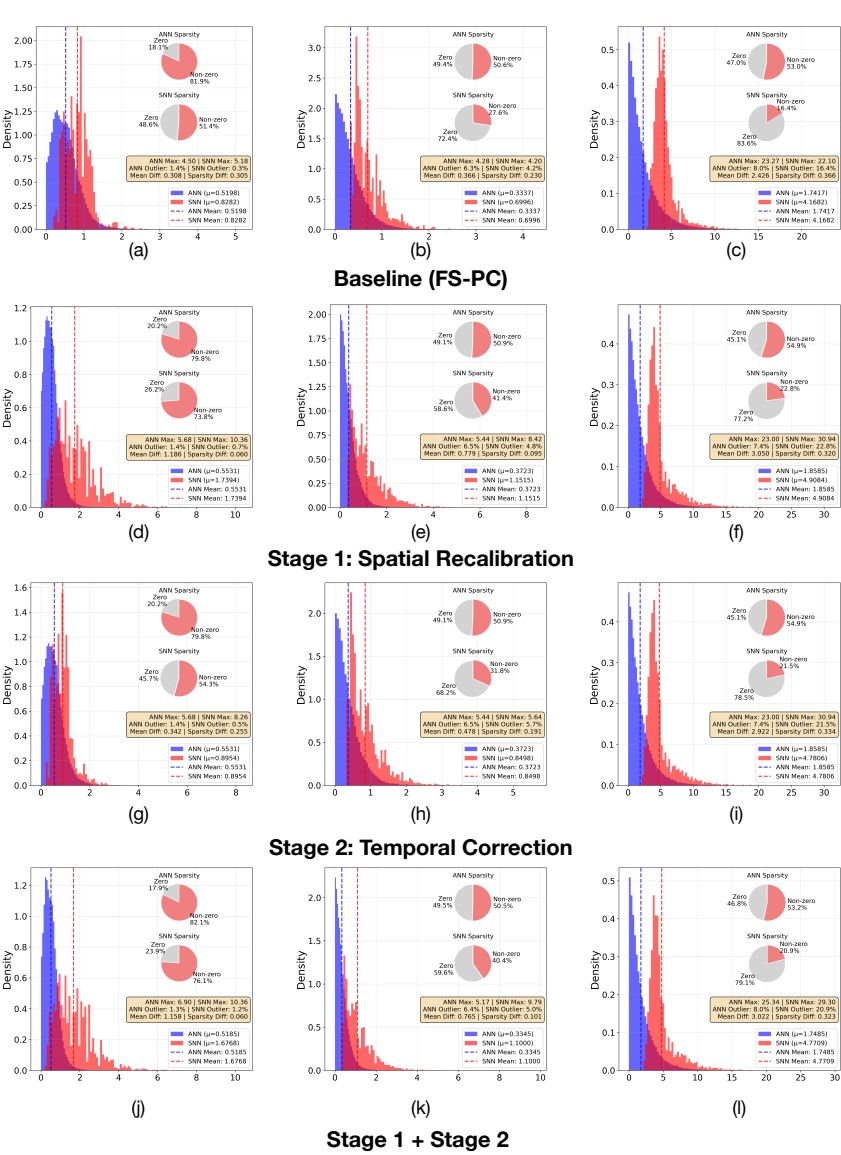

Figure 4: **Visualization of the proposed two-stage calibration on ReLU-based ResNet-18 at** $T = 8$. We compare the output distributions of three representative ReLU / *Parallel IF* layers (2nd, 8th, and 16th/final layer, from left to right) for four configurations: (a-c) Baseline (FS-PC), (d-f) Stage 1: Spatial Recalibration, (g-i) Stage 2: Temporal Correction, and (j-l) Stage 1 + Stage 2 (Our Full Method). Each plot shows the activation value distribution of the source ANN (blue) against the **average firing rate distribution** of the converted SNN (red), along with sparsity and key statistical metrics. Our full two-stage method (j-l) consistently achieves the closest match to the ANN distributions across all layers, minimizing divergence and balancing sparsity.

## C EXPERIMENTAL DETAILS

For Table 1, we use VGG-16 and the widely adopted QCFS-pretrained ResNet-34. For Table 2, we adopt standard ResNet-18/34/50/101 models trained with ReLU activations.

In the QCFS-pretrained setting, activations produced by the quantization–clip–floor–shift (QCFS) operator are converted into parallel spiking neurons for constant-time inference. Entries marked with † employ the calibrated, channel-wise parallel integrate-and-fire variant; non-† models use the plain parallel neuron. For ReLU backbones, we follow this conversion flow: ReLU layers are first replaced by a recorder to capture activation ranges, converted once to QCFS with calibration, and finally substituted by parallel spiking neurons for inference.

**Spiking activations.** All constant-time SNNs use a *parallel integrate-and-fire* (ParaIF) mechanism that generates the entire $T$-step spike train in one pass without recurrent state. The basic **Parallel IF** aggregates the time-averaged input under a descending threshold ladder; our Stage-2 temporal correction adds a lightweight, per-channel bias to this potential. The **Calibrated, channel-wise Parallel IF** extends this with per-channel pre-threshold shifts and post-threshold amplitude adjustments, used for the error-calibrated (†) models. Both remain constant-time and introduce only $10^4$-scale trainable parameters (approximately 7.6k for ResNet-34 and 12.4k for VGG-16). QCFS itself is used only for conversion and calibration, while a non-spiking recorder collects stable activation bounds before replacement.

**Surrogate gradient design.** Spiking nonlinearity is treated as a Heaviside step with a smooth surrogate derivative during fine-tuning. In all our experiments we select the triangle shape, which has unit slope within a small band around the threshold and zero outside.

**Lightweight calibration.** Stage 1 (spatial recalibration) updates only the affine parameters of normalization layers and refreshes running statistics using spiking activations. Stage 2 (temporal correction) optimizes the per-channel bias of the parallel spiking neuron to offset the late-step skew induced by the descending threshold ladder. Both stages use AdamW with a short warm-up and cosine decay; weight decay is omitted. Learning rates are $2 \times 10^{-4}$ for ImageNet (one epoch for Stage 1, Stage 2) and $10^{-3} \to 10^{-4}$ for CIFAR in Stage 1 with $2 \times 10^{-3}$ in Stage 2; gradient clipping is applied in Stage 2.

**Timesteps and batch sizes.** For Table 1, ImageNet experiments use VGG-16 with $T = 2$ and batch size 16, and ResNet-34 with $T = 2$ and batch size 16; CIFAR-10 uses VGG-16 with $T = 2$ and batch size 32; CIFAR-100 uses VGG-16 with $T = 4$ and batch size 64. For Table 2, ImageNet experiments use ResNet-18 with $T = 4$ (batch 16), ResNet-34 with $T = 16$ (batch 128), ResNet-50 with $T = 16$ (batch 32), and ResNet-101 with $T = 16$ (batch 16).

**Algorithm 1** Overall pseudo-code for universal parallel conversion (black: original FS-PC; blue: our added calls)

---

**Require:** Pretrained ANN model $f_{\text{ANN}}$ with $L$ layers; original activation function (QCFS or ClipReLU) $g_{\text{OA}}^l(\cdot)$ and layer-wise output $r_{\text{OA}}^l$; actual activation function (DA-QCFS) $g_{\text{DA}}^l(\cdot)$ and layer-wise output $r_{\text{DA}}^{l,T}$; calibration dataset $\mathcal{D}$; mean function along the channel dimension $\mu(\cdot)$; learning momentum $\alpha$.
**Ensure:** Converted parallel SNN model $f_{\text{SNN}}$.

1: **Stage I: Parameter Initialization**
2: **for** $l = 1$ to $L$ **do**
3:      $f_{\text{ANN}} \cdot g_{\text{DA}}^l \cdot \theta^l \leftarrow f_{\text{ANN}} \cdot g_{\text{OA}}^l \cdot \theta^l$
4:      $f_{\text{ANN}} \cdot \psi_{\text{DA}}^l \leftarrow 0$
5:      $f_{\text{ANN}} \cdot \phi_{\text{DA}}^l \leftarrow 0$
6: **end for**

7: **Stage II: Layer-wise Error Calibration**
8: **for all** (Image, Label) $\in \mathcal{D}$ **do**
9:      **for** $l = 1$ to $L$ **do**
10:         $e_{\text{PRE}}^l \leftarrow \mu\big(W^l r_{\text{OA}}^{(l-1)} - W^l r_{\text{DA}}^{(l-1),T}\big)$
11:         $f_{\text{ANN}} \cdot \psi_{\text{DA}}^l \leftarrow \alpha \cdot f_{\text{ANN}} \cdot \psi_{\text{DA}}^l + (1 - \alpha) \cdot e_{\text{PRE}}^l$

12:         $e_{\text{POST}}^l \leftarrow \mu\big(r_{\text{OA}}^l - r_{\text{DA}}^{l,T}\big)$
13:         $f_{\text{ANN}} \cdot \phi_{\text{DA}}^l \leftarrow \alpha \cdot f_{\text{ANN}} \cdot \phi_{\text{DA}}^l + (1 - \alpha) \cdot e_{\text{POST}}^l$
14:      **end for**
15: **end for**
16: # Ours: call Algorithm 2 on the DA-QCFS-based ANN (after replacing $g_{\text{OA}}^l$ with $g_{\text{DA}}^l$) for BatchNorm affine fine-tuning.

17: **Stage III: Parallel Conversion**
18: **for** $l = 1$ to $L$ **do**
19:      **for** $t = 1$ to $T$ **do**
20:         $f_{\text{SNN}} \cdot b^{l,t} \leftarrow \dfrac{f_{\text{ANN}} \cdot g_{\text{DA}}^l \cdot \theta^l}{2(T - t + 1)} + \dfrac{f_{\text{ANN}} \cdot \psi_{\text{DA}}^l \cdot T}{T - t + 1}$
21:      **end for**
22:      $f_{\text{SNN}} \cdot W^l \leftarrow f_{\text{ANN}} \cdot W^l$
23:      set $f_{\text{SNN}} \cdot \Lambda_{\text{PC}}^l$ according to Eq. (8)
24:      $f_{\text{SNN}} \cdot \theta_{\text{PRE}}^l \leftarrow f_{\text{ANN}} \cdot g_{\text{DA}}^l \cdot \theta^l$
25:      $f_{\text{SNN}} \cdot \theta_{\text{POST}}^l \leftarrow f_{\text{ANN}} \cdot g_{\text{DA}}^l \cdot \theta^l + f_{\text{ANN}} \cdot \phi_{\text{DA}}^l$
26: **end for**

27: # Ours: call Algorithm 3 for output-bias (membrane) fine-tuning on the SNN.

28: **Stage IV: Parallel Inference**
29: **for** $l = 1$ to $L$ **do**
30:      **for** $t = 1$ to $T$ **do**
31:         $I^l \leftarrow f_{\text{SNN}} \cdot W^l \, s^{(l-1)} \, f_{\text{SNN}} \cdot \theta_{\text{POST}}^{(l-1)}$
32:         **if** $f_{\text{SNN}} \cdot \Lambda_{\text{PC}}^l I^l + f_{\text{SNN}} \cdot b^l \geq f_{\text{SNN}} \cdot \theta_{\text{PRE}}^l$ **then**
33:            $s^l \leftarrow 1$                               ▷ fire spikes
34:         **else**
35:            $s^l \leftarrow 0$                               ▷ keep silent
36:         **end if**
37:      **end for**
38: **end for**
39: **return** $f_{\text{SNN}}(W, \Lambda_{\text{PC}}, b, \theta_{\text{PRE}}, \theta_{\text{POST}})$.

# D  THE OVERALL PSEUDO-CODE FOR ANN-TO-SNN PARALLEL CONVERSION

---

**Algorithm 2** BatchNorm affine fine-tuning on DA-QCFS ANN (called in Algorithm 1)

---

**Require:** Calibrated DA-QCFS ANN model $f_{\text{ANN}}$ (after replacing $g_{\text{OA}}^l$ with $g_{\text{DA}}^l$); fine-tune dataset $\mathcal{D}_{\text{train}}$; fine-tune epochs $E_{\text{BN}} = 1$.
**Ensure:** Updated BatchNorm affine parameters and running statistics for $f_{\text{ANN}}$.
 1: Freeze all parameters of $f_{\text{ANN}}$.
 2: Set every BatchNorm layer to train mode and unfreeze only its affine terms (weight and bias).
 3: **for** epoch = 1 to $E_{\text{BN}}$ **do**
 4:     **for all** (Image, Label) $\in \mathcal{D}_{\text{train}}$ **do**
 5:         logits $\leftarrow f_{\text{ANN}}(\text{Image})$                                ▷ no time expansion
 6:         loss $\leftarrow$ CE(logits, Label)
 7:         Update BN affine terms with AdamW using warm-up + cosine learning-rate schedule.
 8:     **end for**
 9: **end for**
10: Set all BatchNorm layers back to eval mode and keep the tuned affine parameters and updated running statistics. =0

---

**Algorithm 3** Output-bias (membrane) fine-tuning on SNN (called in Algorithm 1)

---

**Require:** Parallel-inference SNN $f_{\text{SNN}}$ with ParaInfNeuron layers; fine-tune dataset $\mathcal{D}_{\text{train}}$; simulation length $T$; fine-tune epochs $E_{\text{OB}} = 1$; early-stopping patience $\tau$.
**Ensure:** Updated output-bias (membrane) correction parameters for all ParaInf neurons in $f_{\text{SNN}}$.
 1: Freeze all parameters of $f_{\text{SNN}}$, including BatchNorm.
 2: Collect all ParaInfNeuron output-bias correction parameters
 3: Initialize an AdamW optimizer on the collected parameters with a linear-warmup + cosine-decay learning-rate schedule.
 4: **for** epoch = 1 to $E_{\text{OB}}$ **do**
 5:     **for all** (Image, Label) $\in \mathcal{D}_{\text{train}}$ **do**
 6:         $\text{Image}_T \leftarrow \text{repeat}(\text{Image}, T)$ along the temporal axis and flatten
 7:         $\text{logits}_T \leftarrow f_{\text{SNN}}(\text{Image}_T)$
 8:         $\text{logits} \leftarrow \text{mean}_t(\text{logits}_T)$                         ▷ average over $T$ steps
 9:         loss $\leftarrow$ CE(logits, Label)
10:         Backpropagate loss only through output-bias parameters.
11:         Apply gradient clipping, take one AdamW step, and update the learning-rate scheduler.
12:         **if** early-stopping with patience $\tau$ is triggered **then**
13:             **break**
14:         **end if**
15:     **end for**
16: **end for** =0

---

