# OpenReview forum: "Rethinking the Spatiotemporal Distribution for High-Fidelity Parallel ANN-to-SNN Conversion"
_ICLR.cc/2026/Conference — Submitted to ICLR 2026_

### Official Review · Reviewer_KFXJ · 2025-10-28

**Soundness:** 3
**Presentation:** 2
**Contribution:** 2
**Rating:** 6
**Confidence:** 4

**Summary:**

This work proposes a two-stage distribution-aware calibration method for ANN-SNN Parallel Conversion, which further enhances the performance of SNN parallel inference under the condition of ultra-low time latency.

**Strengths:**

1. This work points out the problem of suppressing early contributions while magnifying late ones in vanilla parallel conversion, and makes exploration based on this.

2. This work provides a theoretical explanation for the LW-FOCF proposed in the second stage.

**Weaknesses:**

1. This work has the most advantages and application value in ultra-low latency and training-free inference scenarios, it actually solves the potential defects of vanilla parallel conversion in correcting quantization errors, as vanilla parallel conversion can already achieve complete lossless conversion for QCFS with $T=\tilde{T}$. Therefore, I tend to think that the main contribution of this work is a local and limited optimization for parallel conversion.

2. Consistent with vanilla parallel conversion, the current applicable visual task and network backbone for this work are still image classification and CNNs. The parallel conversion based on other visual tasks and network structures is worth further consideration.

**Questions:**

See Weaknesses Section.

---

> ### Author Response · Authors · 2025-12-01
> **To Reviewer KFXJ**
>
> We thank the reviewer for the helpful comments.
>
> **W1**:
>
> In practical low-timestep settings (T = 2, 4, or 8) on models such as ResNet-18/34 and VGG-16, which leads to a qualitative degradation of accuracy: performance can drop from around 70\% down to 30-40\%.
>
> Moreover, the "training-free" conversion schemes you refer to in FS-PC[1] still require retraining a custom ANN with a QCFS-like [2] activation function (DA-QCFS in [1]); this retraining is costly. Also, the resulting accuracy is not guaranteed to match that of off-the-shelf pretrained ANN models.
>
> Our method can directly build on off-the-shelf models and turns an almost unusable parallel conversion regime into a performance-competitive and practically deployable one. For example, ResNet-18 (ANN Acc. 69.76\%) from **25.2\% to 62.28\% at T=4**.
>
> Furthermore, even for custom ANN models with QCFS-like activation functions, our approach improves the ResNet-34 accuracy **from 42.45\% to 72.35\% at T=2**.
>
> Overall, our method provides a systematic solution rather than an ad-hoc hack: it rectifies the errors in a **simple**, direct, and unified way, rather than relying on special tricks tailored to a particular network.
>
> **W2**:
>
> Transformer architecture contains nonlinear operators and matrix multiplications (e.g., `Q@K`), and it is still unclear which coding scheme -- rate coding [3], differential coding [4,5], or latency coding [6]  -- is the most suitable.
>
> Under rate coding, the current SOTA method is worse than that of differential coding, and we also expect parallel ANN-to-SNN conversion with rate coding to require more time steps and thus higher energy consumption. For these reasons, we did not include Transformer-based experiments under rate coding. We consider a dedicated validation on SNN-Transformer architectures with rate coding [3] an important direction for future work.
>
> ---
>
> [1] Faster and Stronger -- When ANN-SNN Conversion Meets Parallel Spiking Calculation (ICML 2025)
>
> [2] Optimal ANN-SNN Conversion for High-accuracy and Ultra-low-latency Spiking Neural Networks (ICLR 2022)
>
> [3] Spatio-Temporal Approximation: A Training-Free SNN Conversion for Transformers. (ICLR 2024)
>
> [4] SpikeZIP-TF: Conversion is All You Need for Transformer-based SNN (ICML 2024)
>
> [5] Differential Coding for Training-Free ANN-to-SNN Conversion (ICML 2025)
>
> [6] TTFSFormer: A TTFS-based Lossless Conversion of Spiking Transformer (ICML 2025)

---

### Official Review · Reviewer_HeH2 · 2025-11-01

**Soundness:** 2
**Presentation:** 4
**Contribution:** 2
**Rating:** 4
**Confidence:** 4

**Summary:**

This paper addresses the problem of performance degradation faced by the "parallel conversion" method in the field of ANN-to-SNN conversion under ultra-low time steps. The authors of the paper argue that the reason is the mismatch in spatiotemporal distribution and propose a two-stage calibration method to solve this problem. The improved experimental results demonstrate the effectiveness of the proposed method.

**Strengths:**

1. The writing provides a mathematical proof of the proposed theory, which is logically self-consistent, and the presentation of pictures and tables is appropriate.
2. In terms of results, the accuracy has been improved, demonstrating the effectiveness of the proof method to a certain extent.

**Weaknesses:**

1. Although Appendix B.1 provides ablation results on output distributions across different network depths when Stage 1 or Stage 2 is used independently, it lacks direct ablation experiments quantifying how Stage 1-only or Stage 2-only configurations affect final accuracy. This casts doubt on the effectiveness of Stage 2.
2. A pseudocode description of the algorithm is absent.
3. The paper extensively cites and compares against FS-PC as a baseline. However, the relationship between the proposed method and FS-PC could be more explicitly elaborated. It is recommended that the Methods section clearly specify: (a) the exact components of the FS-PC baseline, and (b) which specific modules the "Ours" method adds to this baseline.

**Questions:**

1. Is the proposed method applicable to converted non-parallel neurons, such as Leaky Integrate-and-Fire (LIF) or Integrate-and-Fire (IF) neurons? This raises uncertainties regarding the method’s scope of application.
2. The authors repeatedly characterize the method as "lightweight," yet the actual costs appear significant relative to the benefits. As indicated in the appendix, Stage 2’s additional calibration step requires approximately 3,000 epochs—even though it introduces only ~10⁴ trainable parameters. Moreover, in the main text section "Stage 2: Ladder-Weighted First-Order Correction Field (LW-FOCF)," the authors note that the training dataset is employed as the calibration set. These substantial additional costs far surpass those associated with retraining an ANN or SNN. We therefore seek clarification: what specific advantages does this method provide in comparison to conventional ANNs?
3. Please clarify the size of the calibration set: In Section 4.1 "MOTIVATION I," the authors state, "Empirical evidence shows that this assumption that pre-BatchNorm activations retain the same distribution breaks down, particularly at ultra-low timesteps T." We request clarification on: (a) What constitutes this "empirical evidence"? (b) The authors assert, "The procedure is lightweight—requiring only a small calibration set and no gradient updates." What is the precise size of this "small calibration set"?

---

> ### Author Response · Authors · 2025-12-01
> **To Reviewer HeH2**
>
> Thank you for the reviewers’ valuable comments.
>
> **W1: Ablations**
>
> We provide accuracy ablation experiments for Stage-1-only and Stage-2-only configurations in the main paper (Table 3, Sec. 5.3), while Appendix B.1 focuses only on visualizing the corresponding layer-wise output distributions. Since the plotted distributions may not show clear separation (stage 1 vs stage 2), we therefore only include the results before and after calibration.
>
> In Table 3, the four columns "Baseline (FS-PC) / Spatial / Temporal / Full" correspond to:
> - Baseline: FS-PC without our calibration,
> - Spatial: Stage 1 only (spatial recalibration),
> - Temporal: Stage 2 only (temporal correction),
> - Full: Stage 1 + Stage 2.
>
> Comparing these columns shows that both Stage 1 and Stage 2 individually improve the final SNN accuracy over the FS-PC baseline, and combining them (“Full”) achieves the best performance.
>
> **W2 and W3**
>
> ```
> Algorithm 1  Universal parallel conversion with two calibration stages
>
> Input:
>     Pretrained ANN f_ANN with L layers and weights W^l;
>     original activation g_OA^l and DA activation g_DA^l;
>     calibration set D and fine-tuning set D_train;
>     simulation length T, momentum α, and optimizer settings.
> Output:
>     Parallel SNN f_SNN.
>
> # Stage I: Initialize DA activations
> for each layer l:
>     copy θ from g_OA^l to g_DA^l;
>     set running correction buffers ψ_DA^l, φ_DA^l to zero.
>
> # Stage II: Layer-wise error calibration (no gradient)
> for (x, y) in D:
>     run f_ANN once with both g_OA^l and g_DA^l;
>     for each layer l:
>         # pre-activation mismatch
>         e_PRE^l  = mean_channel( W^l r_OA^{l−1} − W^l r_DA^{l−1,T} );
>         update ψ_DA^l with momentum α and e_PRE^l;
>         # post-activation mismatch
>         e_POST^l = mean_channel( r_OA^l − r_DA^{l,T} );
>         update φ_DA^l with momentum α and e_POST^l.
>
> # Stage 1(ours): One-epoch BN affine fine-tuning on ANN path
> Freeze all parameters of f_ANN except BatchNorm affine terms (γ, β);
> set all BN layers to train mode.
> for one epoch over D_train:
>     logits = f_ANN(x); loss = CE(logits, y);
>     update only BN affine parameters with AdamW (warm-up + cosine LR).
> Switch BN back to eval mode, keep updated affine and running statistics.
>
> # Stage III: Build parallel SNN parameters
> for each layer l:
>     for timestep t = 1 … T:
>         define time-dependent bias b^{l,t} from θ^l and ψ_DA^l
>         (scale by 1 / (T − t + 1) as in the main text).
>     copy W^l to f_SNN, set Λ_PC^l according to the parallel rule;
>     set θ_PRE^l  = θ^l;
>     set θ_POST^l = θ^l + φ_DA^l.
>
> # Stage 2: One-epoch output-bias fine-tuning on SNN path
> Freeze all SNN weights and BN; make ParaInfNeuron output membrane potential-bias terms learnable.
> for one epoch over D_train:
>     repeat input x along time to length T, run parallel inference;
>     average logits over T, compute CE loss with label y;
>     backprop only through output-bias, apply gradient clipping and early stopping.
> ready for parallel direct inference.
>
> # Stage IV: Parallel inference rule
> for each layer l and timestep t:
>     compute I^l from incoming spikes s^{l−1};
>     if Λ_PC^l I^l + b^{l,t} ≥ θ_PRE^l then
>         emit spike s^l = 1
>     else
>         keep silent s^l = 0.
> ```
>
> **Q1:**
>
> Our current paper is designed for rate-coding SNNs. In this regime, LIF/IF neurons are the standard and simplest neuron models, and therefore we do not anticipate a fundamental restriction on the applicability of our method.
>
> **Q2:**
>
> Clarification on computational cost (typo in Appendix).
> The value "3000" is a max batch steps. In all reported experiments, Stage 2 calibration is performed for only 1 epoch over the calibration set, not 3,000 epochs, which is smaller than retraining from scratch. We will clarify this in the revised version. The calibration dataset can in principle be a subset of the full training set; in our experiments, we chose to run this calibration on the entire training set to pursue the best possible performance.
>
> **Q3:**
>
> By "empirical evidence", we refer to two sources. First, in our own experiments, we explicitly plot the pre-BatchNorm activation distributions under different timesteps T and observe clear shifts as T decreases to the ultra-low regime, contradicting the common assumption that these distributions remain invariant. Second, this phenomenon is also documented in prior work such as Forward Temporal Bias Correction for Optimizing ANN–SNN Conversion (ECCV 2024), which systematically shows that temporal bias and distribution mismatch.
>
> In our original manuscript, the term "gradient-free" was intended only to mean that we do not modify the pretrained backbone parameters during calibration. We realize that this wording is ambiguous. In the revised manuscript, we will rephrase this to make it clear.

---

### Official Review · Reviewer_C4a6 · 2025-11-01

**Soundness:** 1
**Presentation:** 2
**Contribution:** 2
**Rating:** 2
**Confidence:** 5

**Summary:**

The author proposed a calibration pipeline for parallel conversion, which includes updating the BN parameter to match the distribution and introducing a learnable single-channel membrane potential bias to minimize error.  However, However, my concern lies in the practical soundness and the novelty of this method.

**Strengths:**

Improvement over the existing baseline (Hao et al., 2025) and clear figures.

**Weaknesses:**

1. This work focuses on the so called *parallel conversion*. This kind of conversion is not biological plausible that composes all spike train into a single time step. With this design, the SNN will loses its asynchronous advantage. It looks more like a low-bit quantization rather than ANN2SNN conversion. Regarding that, I have additional several concerns:

- The proposed Spatial Recalibration has been a common technique to use in quantization work, such as [1].
- Figure 1 illustrates the distribution mismatch. The authors and prior work claimed a lossless approach under a uniformed distribution prior. But it has been a common sense that activation/parameters distribution never follows a uniform distribution. Therefore, the behavior shown in Fig.1 is in line with everyone's expectations.
- Furthermore, the distribution mismatch problem is not unique problem in the parallel conversion. Even in standard case, the ReLU2LIF conversion introduces the mismatch. This problem has been clearly illustrated in SNN Calibration proposed 4 years ago[2], which also proposed similar calibration techniques.

2.  I have concerns about the scope of this work. It seems that this work primarily focused on the ResNet-34 and VGG-16 on the ImageNet dataset. However, even a much tiny architecture like MobileViT-S can achieve 78.4 accuracy with 5.6M parameters. I wonder how this work can be used to prove the SNN efficiency combined with latest ANN advancements.

3.  The goal of this work is to obtain extremely low timesteps SNNs. Can the authors analyze which one is better, the Quantized ANN with 2-bit activations or the Converted SNN with 4 timesteps？My opinion is 2-bit quantized ANN is better given by its less activation memory and easier hardware logic.

[1] PD-Quant: Post-Training Quantization based on Prediction Difference Metric. CVPR 2023

[2] A Free Lunch From ANN: Towards Efficient, Accurate Spiking Neural Networks Calibration. ICML 2021

**Questions:**

See weakness. Besides, for point 1, how does the author view the similarity with [2] and exising spatial recalibration techniques, and what is the essential difference between the contribution of this paper and directly applying those existing recalibration techniques to SNNs with parallel neurons?

---

> ### Author Response · Authors · 2025-11-30
> **To Reviewer C4a6**
>
> We thank the reviewer for the above comments.
>
> (1) On “biological plausibility” and the loss of asynchronous advantages.
> We would like to clarify two points. First, compressing spike activity into short, intense windows is not entirely foreign to neuroscience: burst-based coding has long been viewed as a meaningful information unit [1,2,3]. We only use this as a loose analogy, not as a strict biological realism. From a practical standpoint, neuromorphic chips are not required to be designed to strictly compute in the same way as the human brain. After all, the brain’s computation speed and capacity are relatively limited. As long as the assumptions remain reasonable, exceeding biological capabilities is acceptable.
>
> Second, “parallel” in our setting does not eliminate asynchrony at the system level. Although the spikes of one layer are time-collapsed for each sample, different layers and different samples can still be scheduled in an asynchronous pipeline. In other words, the hardware can still exploit event-driven sparsity and layer-wise pipelining; our method changes the coding scheme, not the fact that spikes are processed in an event-driven manner.
>
> ```
>                                        time
>         --------------------------------------------------------------->
>
>          →  →  →  → +---------+
> layer1  |a4|a3|a2|a1|    1    |
>         |           +------+--+
>         |                  |b1|
>         |                  |b2|
>         |                  |b3|
>         |                  |b4|
>         |                  +--+
>         |                    →+---------+
> layer2  |                    →|    2    |
>         |                    →+------+--+
>         |                            |c1|
>         |                            |c2|
>         |                            |c3|
>         |                            |c4|
>         |                            +--+
>         |                              →+---------+
> layer3  |                              →|    3    |
>         |                              →+------+--+
>         |                                      |d1|
>         |                                      |d2|
>         |                                      |d3|
>         |                                      |d4|
>         |                                      +--+
>         |
>         |          →         →         →
>         |→ +---------+---------+---------+---------+
> layer1: |a1|    1  a2|    1  a3|    1  a4|    1    |
>            +---------+---------+---------+---------+
>                              →         →         →
>                    → +---------+---------+---------+---------+
> layer2:           |b1|    2  b2|    2  b3|    2  b4|    2    |
>                      +---------+---------+---------+---------+
>                                        →         →         →
>                              → +---------+---------+---------+---------+
> layer3:                      c1|    3  c2|    3  c3|    3  c4|    3    |
>                                +---------+---------+---------+---------+
>
> ```
>
> (2) Regarding quantization vs SNN.
>
> There is indeed a superficial similarity: both our parallel SNN and a low-bit ANN can be seen as operating on discretized activations. However, they are not equivalent.
>
> A low-bit quantized ANN is stateless: the activation at each layer is a pure function of the current input. In contrast, our parallel SNN maintains membrane states and allows accuracy–latency trade-offs by increasing the effective number of timesteps. In practice we observe that increasing T improves accuracy in a way that cannot be reproduced by simply re-evaluating a quantized ANN multiple times. The design is essentially developed from a time-step perspective, for example by correcting the temporal and spatial distributions at different time steps.
>
> For the other points, please refer to my response to Reviewer `a7GX` about Weakness 1 and 2.
>
> ---
>
> [1]: Dendritic localized learning: Toward biologically plausible algorithm. (ICML 2025)
>
> [2]: Bursts as a unit of neural information: making unreliable synapses reliable (Trends Neurosci, 1997)
>
> [3]: Neural Coding With Bursts (Zeldenrust et al., 2018)

---

### Official Review · Reviewer_a7GX · 2025-11-08

**Soundness:** 2
**Presentation:** 3
**Contribution:** 3
**Rating:** 4
**Confidence:** 5

**Summary:**

This paper investigates the severe accuracy degradation of parallel ANN-to-SNN conversion under ultra-low timesteps T, attributing it to a spatiotemporal distributional mismatch: spatially, BatchNorm statistics are misaligned with spike-domain activations; temporally, the parallel firing rule concentrates spikes in late timesteps. The authors propose a two-stage calibration framework spatial recalibration and temporal bias correction, that significantly boosts ImageNet accuracy for ResNet-18/34 especially at T=4/8.

**Strengths:**

1.The analysis of spatiotemporal bias in parallel conversion is clear , particularly the breakdown of QCFS assumptions at ultra-low T.

2.The reported improvement on ImageNet is competitive, such as ResNet-18 from 25.2% to 62.28% at T=4.

**Weaknesses:**

1.Both stages essentially combine existing ideas ,which is lack of innovation.

2.Experiments are restricted to CNNs on image classification benchmarks. The method is not evaluated on modern Transformer architectures, which dominate current vision and language tasks.

3.A core motivation for SNNs is ultra-low power consumption, yet the paper provides no energy estimates, spike activity statistics, or hardware deployment analysis

**Questions:**

1.Is the temporal bias dependent on a fixed T? If T changes (e.g., from 4 to 8), must the model be recalibrated?

---

> ### Author Response · Authors · 2025-11-30
> **To Reviewer a7GX about Weakness 1**
>
> _I would like to thank the reviewer for the concrete comments and valuable suggestions._
>
> ### W1
> Let me also clarify the **innovation** of our work and how it **differs from prior studies**. We focus on **parallel ANN-to-SNN conversion**, with FS-PC [1] as our main baseline. FS-PC is the **first** paper that really opens up this direction, and our contribution is to address a core practical issue within **this specific setting**.
>
> In FS-PC, to get good performance one needs to **retrain** a customized ANN [3] for the parallel setting or use many timesteps, which is expensive and hurts usability. **Our contribution** is very **pragmatic**: with a **simple** modification, we **enable off-the-shelf ANNs** to be converted into parallel SNNs that achieve **strong performance** at low timesteps without retraining. We hope the reviewer can evaluate our contribution under this concrete background, as turning a barely usable setting into one with a clear qualitative **accuracy improvement**.
>
> If we correct the model from the **spatial perspective**, then we must consider, in particular, correcting the **data distribution**. This is a very natural consideration. [1] has already moved in this direction which introduces a distribution-aware QCFS (DA-QCFS) module that learns a per-channel shift $\psi_{\text{DA}}^l$ and scale $ \phi_{\text{DA}}^l $ by greedily minimizing the mean conversion error before and after each activation on a small calibration set. Even DA-QCFS behaves very much like a heuristic normalization-like affine correction, it fails for off-the-shelf standard ANN model. We correct the feature distribution by finetuning the normalization layers directly, which is also a very **natural, simple, and direct choice**: normalization is already built into the model, so why not just use it directly? **Simplicity** in the method is also important. A simple method is not a bad method. We would not mind inserting additional **normalization layers** at multiple locations in the model to further correct the error.
>
> On the **temporal side**, most **prior works** rely on heuristic tuning of initial membrane potentials [4], thresholds, offsets spikes, or bias shifts [2]. FS-PC [1] and FTBC [2] also **adjust input/output biases**, yet still struggle to obtain high accuracy when converting an off-the-shelf ANN to a parallel SNN at low timesteps. Our perspective is that every timestep contributes error, so in the parallel setting it is natural to aggregate these timestep-dependent errors into a time-collapsed membrane-potential bias. We turn output bias correction into parameters that can be globally fine-tuned. After only a few epochs, this yields significant performance gains while keeping the method simple and easy to use.
>
> To **summarize**, there have indeed been prior attempts, but they mainly target corrections for **non-parallel** ANN-to-SNN conversion. Even when prior work tries to adjust the feature distribution in the parallel setting, for example by using normalization or bias terms, the **performance is still not satisfactory**. Our work takes a pragmatic step forward and achieves clear performance improvements, and we hope the reviewer can recognize this contribution.
>
> ---
>
> [1] Faster and Stronger – When ANN-SNN Conversion Meets Parallel Spiking Calculation (ICML 2025)
>
> [2] Forward Temporal Bias Correction for Optimizing ANN-SNN Conversion (ECCV 2024)
>
> [3] Optimal ANN-SNN Conversion for High-accuracy and Ultra-low-latency Spiking Neural Networks (ICLR 2022)
>
> [4] Optimized Potential Initialization for Low-Latency Spiking Neural Networks (AAAI 2022)

---

> > ### Author Response · Authors · 2025-11-30
> > **To Reviewer a7GX about Weakness 2, 3 and Question 1**
> >
> > ### W2, W3
> >
> > Modern Transformer architectures contain many nonlinear operators and Q@K matrix multiplications, and it is still unclear which coding scheme—rate coding [6], latency coding [7], or differential coding [5,8]—is the most suitable in this setting. Under rate coding, the current SOTA method is worse than that of differential coding, and we also expect parallel ANN-to-SNN conversion with rate coding to require more time steps and thus higher energy consumption. For these reasons, we did not include Transformer-based experiments under rate coding. We consider a dedicated validation on SNN-Transformer architectures an important direction for future work.
> >
> > For rate coding, it is well understood that the closer the firing rate is to the ANN activation value, the higher the resulting SNN accuracy tends to be. A substantial body of prior work has already employed the energy estimation model in [10] to argue that SNNs can be significantly more energy-efficient than conventional ANNs. Building on these results, we can roughly approximate the energy consumption in our setting by referencing related work [9]. For example, SNN-VGG-16 with T=4 is reported to consume around 10 mJ [10], which is approximately 10\% of the energy used by its ANN counterpart [11]. But energy estimation in our community is still coarse, and several papers have explicitly highlighted this limitation [12, 13].
> >
> >
> >
> > [5] SpikeZIP-TF: Conversion is All You Need for Transformer-based SNN (ICML 2024)
> >
> > [6] Spatio-Temporal Approximation: A Training-Free SNN Conversion for Transformers. (ICLR 2024)
> >
> > [7] TTFSFormer: A TTFS-based Lossless Conversion of Spiking Transformer (ICML 2025)
> >
> > [8] Differential Coding for Training-Free ANN-to-SNN Conversion (ICML 2025)
> >
> > [9] Temporal Misalignment in ANN-SNN Conversion and Its Mitigation via Probabilistic Spiking Neurons (ICML 2025)
> >
> > [10] Computing’s energy problem and what we can do about it (ISSCC 2014)
> >
> > [11] Rethinking skip connections in Spiking Neural Networks with Time-To-First-Spike coding, https://arxiv.org/abs/2312.00919
> >
> > [12] Reconsidering the energy efficiency of spiking neural networks,
> > https://arxiv.org/pdf/2409.08290v1
> >
> > [13] Are SNNs Truly Energy-efficient? -- A Hardware Perspective, https://arxiv.org/pdf/2309.03388
> >
> > ---
> >
> > ### Q1
> >
> > In Table 1, we calibrate the model separately for T=2 and T=4. However, we also observe that if we calibrate the model directly at T=4, its intermediate time step t=2 already achieves higher accuracy than the model calibrated explicitly for T=2.
> >
> > **ResNet-34 Acc. (\%)**
> >
> > | Method  | t   | 1     | 2        | 3     | 4        |
> > |--------|-----|-------|----------|-------|----------|
> > | FS-PC  | T=2 | -     | 42.45    | -     | -        |
> > | FS-PC  | T=4 | -     | -        | -     | 67.28    |
> > | Ours   | T=2 | 67.56 | 68.41    | -     | -        |
> > | Ours   | T=4 | 70.72 | **72.35**| 72.94 | **73.24**|
> > | FS-PC †| T=2 | -     | 65.20    | -     | -        |
> > | FS-PC †| T=4 | -     | -        | -     | 72.90    |
> > | Ours † | T=2 | 68.29 | 69.27    | -     | -        |
> > | Ours † | T=4 | 70.68 | **72.32**| 72.94 | **73.10**|
> >
> >
> > **VGG-16 Acc. (\%)**
> >
> > | Method  | t   | 1     | 2        | 3     | 4        |
> > |--------|-----|-------|----------|-------|----------|
> > | FS-PC  | T=2 | -     | 36.98    | -     | -        |
> > | FS-PC  | T=4 | -     | -        | -     | 71.23    |
> > | Ours   | T=2 | 53.21 | 63.91    | -     | -        |
> > | Ours   | T=4 | 50.85 | **65.19**| 70.63 | **72.23**|
> > | FS-PC †| T=2 | -     | 56.50    | -     | -        |
> > | FS-PC †| T=4 | -     | -        | -     | 71.75    |
> > | Ours † | T=2 | 49.18 | 61.84    | -     | -        |
> > | Ours † | T=4 | 53.09 | **66.35**| 71.28 | **72.65**|

---

### Author Response · Authors · 2025-12-03
**To PC, SAC, and AC**

> “We can only see a short distance ahead, but we can see plenty there that needs to be done.”
> -- Alan Turing

To PC, SAC, and AC,

Thank you for handling our submission and for the detailed reviews and discussion. We have carefully revised the paper and would like to briefly restate what this work contributes, and how the revision addresses the core concerns.

1. Problem and scope.
   We focus specifically on **parallel ANN-to-SNN conversion at ultra-low timesteps** (e.g., $T \le 4$) for **off-the-shelf ReLU-based ImageNet backbones** (ResNet-18/34, VGG-16), as well as QCFS/DA-QCFS–trained backbones. Existing parallel methods work reasonably well only when the ANN has been retrained with QCFS-like activations; off-the-shelf ReLU models remain severely under-explored and perform poorly in this regime.

2. Main technical idea.
   We show that parallel conversion suffers from a **spatiotemporal distribution mismatch**:
   – Spatially, BatchNorm layers operate on ANN statistics that no longer match spike-domain activations.
   – Temporally, the QCFS-derived **time-dependent bias** in the parallel firing rule induces a systematic skew.
   To fix this, we propose a **two-stage, low-cost calibration** that keeps the backbone and parallel rule unchanged:
   – **Stage 1 (Spatial):** bring the backbone into a DA-QCFS form and recalibrate BN running statistics and affine terms under this DA-QCFS rate proxy, so that normalized features match the firing-rate targets used in parallel conversion.
   – **Stage 2 (Temporal):** treat the parallel neuron as an analytically derived time-dependent neuron from IF + QCFS, and add a **lightweight per-channel bias correction** on top of the QCFS-derived bias to cancel the timestep-dependent skew.

3. Empirical impact.

    On ImageNet, starting from **off-the-shelf ReLU backbones**, our calibration turns previously unusable ultra-low $T$ regimes into practical ones, e.g.:
   – ResNet-18, $T=4$: **25.20% → 62.28%**;
   – ResNet-34, $T=8$: **50.67% → 68.23%**.

    On **QCFS/DA-QCFS–trained backbones**, we further improve the strongest parallel baseline:
   – ResNet-34, $T=2$: **42.45% → 72.35%**;
   – VGG-16, $T=2$: **36.98% → 65.19%**.
   We also include systematic ablations (spatial-only, temporal-only, full) and intermediate-timestep evaluations that clarify where the gains come from.

4. Clarifications vs. prior work and “just quantization” concerns.

The revised paper now **explicitly derives** the parallel neuron coefficients from IF dynamics and QCFS, and contrasts them with generic activation quantization in an appendix. We clarify that our method does **not** introduce arbitrary quantization parameters: it starts from the **QCFS-derived time-dependent bias**. We also streamline the theory, remove unnecessary “theorem-style” packaging, and rewrite the Method section to be more transparent and concrete.

We hope this clarifies the intention and scope of the work. Below, we respond to each reviewer’s comments point by point and indicate the corresponding changes in the revised manuscript.

Thank you for your time.

Sincerely,

Authors

---

### Meta-Review · Area_Chair_5muy · 2025-12-19

**Summary:**

This paper proposes a two-stage calibration framework (spatial recalibration and temporal bias correction) to address the severe accuracy degradation of parallel ANN-to-SNN conversion at ultra-low timesteps (T=2,4,8). Reviewers acknowledge the clear problem identification, solid empirical improvements on ImageNet with CNNs (e.g., ResNet-18/34, VGG-16), and improved presentation in the revision. However, the reviewers converge on a critical and fundamental weakness: the core methodological contribution is perceived as incremental and lacking in novelty. The proposed calibration stages are seen as straightforward combinations or applications of existing techniques (e.g., BatchNorm fine-tuning, bias correction) from related fields like quantization and prior SNN calibration works, adapted to the specific parallel conversion setting. While the performance gains are notable, they are viewed as a pragmatic engineering improvement rather than a conceptual or theoretical advance that significantly pushes the field forward.

**Reviewer Concerns:**

Addressed/Clarified:

Ablation Study Clarity (HeH2): Authors provided accuracy results for Stage-1-only and Stage-2-only configurations (Table 3) and added a pseudocode algorithm.

Clarification vs. Baseline (HeH2): The relationship with the FS-PC baseline and the specific added components are now clearer.

Computational Cost Clarification (HeH2): Authors corrected a misleading description about calibration epochs (1 epoch, not 3000).

Outstanding Core Concerns (Leading to Rejection):

Insufficient Innovation / Incremental Nature (a7GX, C4a6, KFXJ): This is the most consistent and damning critique.

Spatial Recalibration is viewed as a direct application of calibration techniques common in post-training quantization (C4a6).

Temporal Bias Correction is seen as a logical extension of known temporal mismatch issues, similar to ideas in prior SNN calibration works like SNN Calibration (ICML 2021) (C4a6).

The integration of these two stages is considered a straightforward combination rather than a novel synergy. As Reviewer a7GX states, both stages "essentially combine existing ideas." Reviewer KFXJ frames it as a "local and limited optimization."

Blurred Line with Quantization & Questionable SNN Contribution (C4a6): A profound concern questions the very nature of the contribution. The "parallel conversion" scheme compresses spikes into a single or few timesteps, leading a reviewer to argue it "looks more like a low-bit quantization rather than ANN2SNN conversion" and loses the asynchronous advantage central to SNNs. The authors' rebuttal draws an analogy to burst coding but does not convincingly counter the critique that the work's essence is efficient discretization/quantization rather than a novel SNN mechanism.

Narrow Scope and Limited Generalizability (a7GX, C4a6, KFXJ): The evaluation is confined to CNNs on ImageNet. Reviewers explicitly note the lack of validation on modern architectures (e.g., Transformers) which are dominant in current vision research, limiting the perceived impact and generality of the claimed "universal" method.

Lack of Energy Efficiency Analysis (a7GX): A key motivation for SNNs is low-power inference, yet the paper provides no energy estimates or spike activity analysis to substantiate the practical efficiency claims of the calibrated parallel SNNs versus quantized ANNs.

**Reviewer Scores:**

Reviewer a7GX (Initial: 4): Explicitly cites "lack of innovation" as a weakness. The authors' rebuttal defends practicality but does not address the novelty concern. The discussion with other reviewers (especially C4a6) reinforcing this point would likely solidify their view. Final score would likely remain at 4 or decrease to 3.

Reviewer C4a6 (Initial: 2, Reject): Holds the most severe view, questioning the fundamental contribution and alignment with SNN research. The authors' response is unlikely to change this assessment. Final score would firmly remain at 2 (Reject).

Reviewer HeH2 (Initial: 4): Concerns were largely clarificatory and addressed. However, in a discussion highlighting the pervasive innovation critique from other reviewers, they might reconsider the contribution score. Final score might remain at 4 or lower to 3.

Reviewer KFXJ (Initial: 6): Already frames the contribution as a "local and limited optimization." In a discussion where the lack of broad novelty and scope becomes the central theme, this reviewer might acknowledge the work's incremental nature relative to conference standards. Final score would likely lower to 4.

---

### Decision · Program_Chairs · 2026-01-26

Reject